# The Arabidopsis SHORTROOT network coordinates shoot apical meristem development with auxin-dependent lateral organ initiation

**Elmehdi Bahafid[1], Imke Bradtmöller[1], Ann M Thies[1], Thi TON Nguyen[1], Crisanto Gutierrez[2], Bénédicte Desvoyes[2], Yvonne Stahl[1], Ikram Blilou[3], Rüdiger GW Simon[1]***

[1]Institute for Developmental Genetics, Heinrich Heine University, Düsseldorf, Germany; [2]Centro de Biologia Molecular Severo Ochoa, CSIC-UAM, Cantoblanco, Madrid, Spain; [3]Laboratory of Plant Cell and Developmental Biology, Division of Biological and Environmental Sciences and Engineering, 4700 King Abdullah University of Science and Technology, Thuwal, Saudi Arabia

*For correspondence:
ruediger.simon@hhu.de

Competing interest: The authors declare that no competing interests exist.

**Abstract** Plants produce new organs post-embryonically throughout their entire life cycle. This is due to stem cells present in the shoot and root apical meristems, the SAM and RAM, respectively. In the SAM, stem cells are located in the central zone where they divide slowly. Stem cell daughters are displaced laterally and enter the peripheral zone, where their mitotic activity increases and lateral organ primordia are formed. How the spatial arrangement of these different domains is initiated and controlled during SAM growth and development, and how sites of lateral organ primordia are determined in the peripheral zone is not yet completely understood. We found that the SHORT-ROOT (SHR) transcription factor together with its target transcription factors SCARECROW (SCR), SCARECROW-LIKE23 (SCL23) and JACKDAW (JKD), promotes formation of lateral organs and controls shoot meristem size. SHR, SCR, SCL23, and JKD are expressed in distinct, but partially overlapping patterns in the SAM. They can physically interact and activate expression of key cell cycle regulators such as *CYCLIND6;1* (*CYCD6;1*) to promote the formation of new cell layers. In the peripheral zone, auxin accumulates at sites of lateral organ primordia initiation and activates SHR expression via the auxin response factor MONOPTEROS (MP) and auxin response elements in the *SHR* promoter. In the central zone, the SHR-target SCL23 physically interacts with the key stem cell regulator WUSCHEL (WUS) to promote stem cell fate. Both SCL23 and WUS expression are subject to negative feedback regulation from stem cells through the CLAVATA signaling pathway. Together, our findings illustrate how SHR-dependent transcription factor complexes act in different domains of the shoot meristem to mediate cell division and auxin dependent organ initiation in the peripheral zone, and coordinate this activity with stem cell maintenance in the central zone of the SAM.

## Editor's evaluation

This is a valuable study of Arabidopsis shoot apical meristem maintenance and organ initiation, with a focus on how SHR, SCR, JKD, and SCL23, four transcription factors initially characterized in root meristems, are deployed in a different context. The imaging, genetics and FRET-FLIM evidence supporting the claims of the authors is compelling. The work will be of interest and importance for plant developmental biologists.

## Introduction

In multicellular organisms, stem cells are the source of all tissues and organs. In contrast to animals where organs are formed during embryogenesis, plants form organs postembryonically, and they can continuously produce new organs throughout their life span. This process depends on the activity of pluripotent stem cells embedded in their shoot and root apical meristems (SAM and RAM, respectively).

In the SAM, stem cells are located in the central zone at the tip of the meristem. Directly underneath lies the organizing center, which is required for stem cell maintenance. The stem cells in the central zone have a low mitotic activity, but after stem cell division, descendants are displaced towards the peripheral zone. Here, they start to divide faster and provide founder cells for lateral organ primordia that will develop at the meristem flank (*Reddy et al., 2004*). How the communication between these distinct functional domains is coordinated to balance stem cell proliferation in the center with organ formation at the periphery of the SAM is still not understood. Studies on the root apical meristem (RAM) showed how mobile transcription factors and their interaction in locally formed protein complexes can maintain stem cell niches and formation of new tissue layers. In the RAM, the GRAS family transcription factor (TF) SHORT-ROOT (SHR) is transcribed in the stele, but SHR protein moves one cell layer outward to interact with the GRAS domain TFs SCARECROW (SCR) and potentially SCARECROW LIKE 23 (SCL23), and with the BIRD/INDETERMINATE DOMAIN TF JACKDAW (JKD; *Long et al., 2015a*; *Long et al., 2017*; *Nakajima et al., 2001*; *Cui et al., 2007*). The resulting multimeric complex promotes asymmetric cell division (ACD) of stem cells, that is the Cortex-Endodermis Initial/daughter (CEI/CEID), by activating expression of the cell cycle regulator *CYCLIND6;1* (*CYCD6;1*) to instruct development of distinct endodermis and cortex cell layers (*Long et al., 2015b*; *Long et al., 2015a*; *Long et al., 2017*; *Di Laurenzio et al., 1996*; *Helariutta et al., 2000*; *Nakajima et al., 2001*). SHR and SCR are required for the maintenance and specification of the quiescent center, as *shr* and *scr* mutants show abnormal quiescent center cells and roots remain short (*Sabatini et al., 2003*; *Petersson et al., 2009*). We now asked whether stem cell maintenance and tissue patterning in the SAM involves the SHR signaling pathway.

In the SAM, stem cell maintenance depends on the homeodomain transcription factor WUSCHEL (WUS) (*Laux et al., 1996*; *Zhang et al., 2017*). *WUS* transcripts are limited to the organizing center, but WUS protein moves via plasmodesmata from the organizing center to the central zone to directly activate the transcription of the secreted signalling peptide CLAVATA3 (CLV3) in stem cells (*Laux et al., 1996*; *Yadav et al., 2011*; *Daum et al., 2014*). In turn, CLV3 acts to limit *WUS* expression to the organizing center via a system of receptor-like kinases and co-receptors: CLAVATA1 (CLV1), CLAVATA2 (CLV2), CORYNE (CRN) and BARELY ANY MERISTEM1-3 (BAM1-3) (*Bleckmann and Simon, 2009*; *Clark et al., 1997*; *Fletcher et al., 1999*; *Nimchuk et al., 2015*; *Ohyama et al., 2009*). The resulting WUS-CLV3 negative feedback circuit controls stem cell homeostasis in the SAM (*Brand et al., 2000*; *Schoof et al., 2000*). Recently it was shown that WUS interacts in the organizing center with the GRAS family transcription factors HAIRY MERISTEM 1 and 2 (HAM1/2) to restrict CLV3 activation to the apical domain of the central zone (*Han et al., 2020b*). Formation of lateral organ primordia at the meristem flanks is preceded by establishment of a local maximum of auxin (*Benková et al., 2003*; *Heisler et al., 2005*; *Vernoux et al., 2011*), which is generated by the auxin efflux carrier PIN-FORMED1 (PIN1) and polar auxin transport (*Friml et al., 2004*; *Gälweiler et al., 1998*). The transcriptional auxin read-out involves the Auxin/Indole-3-acetic acid (Aux/IAA) repressor proteins that under low auxin conditions dimerize with auxin response factors (ARFs) and repress their activity (*Mockaitis and Estelle, 2008*). ARFs bind DNA at the promoters of their target genes via an auxin response element (AuxRE; *Boer et al., 2014*). An increase in auxin levels triggers the formation of a complex between Aux/IAA and TIR1/AFB leading to the degradation of Aux/IAA and the release of the ARFs, which can now regulate expression of auxin response genes (*Paque and Weijers, 2016*). Auxin-dependent lateral organ initiation is mediated by AUXIN RESPONSE FACTOR 5/MONOPTEROS (ARF5/MP; *Berleth and Jürgens, 1993*; *Hardtke and Berleth, 1998*). MP plays a crucial role in the initiation of flower primordia as the strong *mp* allele lacks roots and a weak allele cannot produce flowers and forms a naked inflorescence stem (*Yamaguchi et al., 2013*). MP directly induces the expression of its target gene *LEAFY (LFY)*, a plant-specific transcription factor that plays a key role in flower primordia initiation and in primordium fate specification (*Schultz and Haughn, 1991*; *Weigel et al., 1992*; *Yamaguchi et al., 2013*; *Wu et al., 2015*).

Here, we started to unravel how organ initiation at the periphery is coordinated with stem cell behavior in the center of the SAM. We show that SHR, SCR, SCL23, and JKD act in the SAM in different expression domains with complementary and overlapping patterns, and together, they cover all the functional domains of the SAM. Interestingly, the overlapping region of the four TFs coincides with the expression domain of *CYCD6;1*. We find that mutants in the *SHR* gene regulatory network show reduced meristem sizes and delayed cell cycle progression due to an extended G1 phase. We demonstrate that auxin via MP promotes expression of *SHR* and *SCR* in the peripheral zone, where *SHR* acts upstream of *LFY* to promote flower primordia formation. In the meristem center, members of the SHR network directly interact with the WUS-CLV circuit. Together, our study shows how mobile TFs of the SHR network establish a balance between stem cell maintenance and lateral organ initiation, by controlling cell division rates and cell fate in different domains within the SAM in an auxin dependent manner.

## Results

### *SHR* and *SCR* coordinate shoot meristem size with primordia initiation and auxin signaling in the SAM

To investigate the roles of *SHR* and *SCR* in shoot meristem development, we analyzed the phenotypes of *shr-2*, *scr-3*, and *scr-4* mutants. All mutants displayed a small rosette, dwarfed shoot phenotype and initiated fewer flowers compared to wild type (WT; *Figure 1A–C*; *Figure 1—figure supplement 1A–F*). The interval between the initiation of successive lateral organs, the plastochron, was increased in the mutants (*shr-2*, *scr-3*, and *scr-4*) compared to WT, indicating a significant delay in lateral organ primordia initiation (*Figure 1D*; *Figure 1—figure supplement 1A'-D' and G*). Using confocal microscopy, we measured meristems of mutants and WT (*Figure 1E-G*; *Figure 1—figure supplement 1I and J*) and calculated the surface area of the SAM excluding organ primordia. Mutants developed significantly smaller meristems (*Figure 1H*; *Figure 1—figure supplement 1H*), suggesting that SHR and SCR might coordinate meristem size with lateral organ primordia initiation during SAM development. Average cell sizes showed no differences between mutants and WT (*Figure 1—figure supplement 1K*), but cell numbers were reduced in all mutants (*Figure 1L*; *Figure 1—figure supplement 1L*). We therefore examined cell proliferation rates using the plant cell cycle marker PlaCCI (*Desvoyes et al., 2020*; *Figure 1I–K*; *Figure 1—figure supplement 1M and N*), which marks the G1, S+G2, and mitotic (M) phases with reporters *pCDT1a:CDT1a-eCFP*, *pHTR13:HTR13-mCherry*, and *pCYCB1;1:N-CYCB1;1-YFP*, respectively. The percentage of G1-marked cells with CDT1a-eCFP was significantly higher in the SAMs of *shr-2*, *scr-3*, and *scr-4* mutants compared to WT (*Figure 1P*; *Figure 1—figure supplement 1O and P*), indicating delayed progression through the cell cycle. Our findings suggest that *SHR* and *SCR* promote cell cycle progression in the SAM by controlling the G1 phase, potentially contributing to the reduced SAM size and delayed organ initiation in the PZ of *shr* and *scr* mutants (*Figure 1D and H*; *Figure 1—figure supplement 1G and H*). Because organ initiation sites are determined by auxin accumulation and signaling (*Vernoux et al., 2011*), we investigated whether loss of *SHR* or *SCR* functions affected auxin levels, which could explain the delayed lateral organ primordia initiation in *shr* and *scr* mutants (*Figure 1D*; *Figure 1—figure supplement 1G* ). We analysed the auxin transcriptional output reporter *pDR5v2:3xYFP-N7* (*Heisler et al., 2005*) in the *shr-2* and *scr-4* mutant backgrounds (*Figure 1M–O*, arrowheads), and found that average number of DR5-positive domains in the SAMs of *shr-2* and *scr-4* mutants was significantly reduced (*Figure 1T*), indicating that SHR and SCR are required for establishing auxin output maxima in the SAM. Using the auxin input sensor R2D2 (*Liao et al., 2015*), which reports auxin levels based on the fluorescence ratio between Venus and tdTomato, we found that low auxin areas expanded in the meristems of *shr-2* and *scr-4* mutants, sometimes beyond lateral organ boundaries into lateral organ primordia (*Figure 1—figure supplement 1Q–S*, arrowheads). These results indicate that *SHR* and *SCR* play crucial roles in maintaining normal auxin distribution within the SAM, which is essential for the generation of defined auxin maxima in the peripheral zone and subsequent lateral organ primordia initiation.

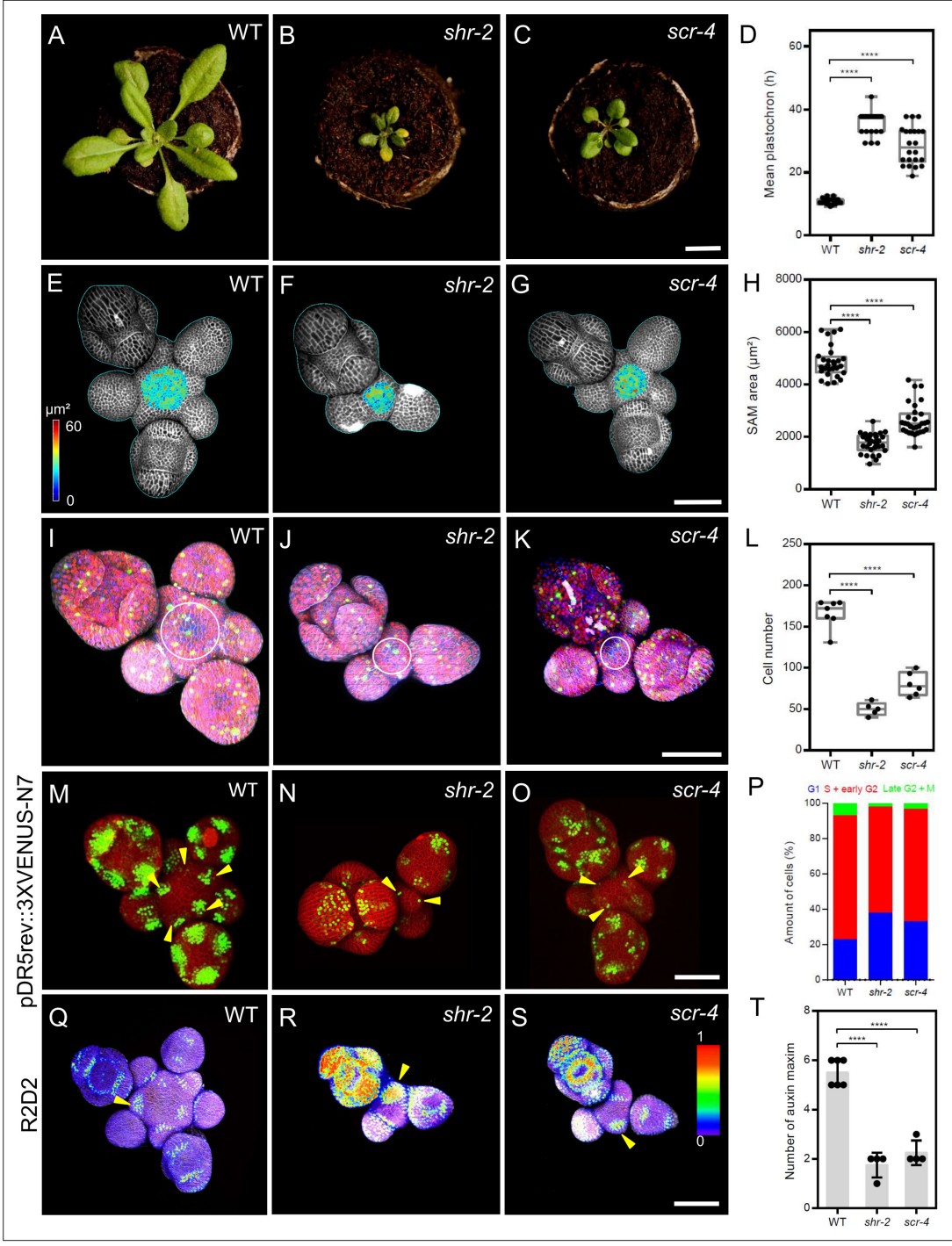

**Figure 1.** SHR and SCR functions modulate meristem size and auxin signalling in the shoot apical meristem. (**A–C**) Top view of 21-day-old rosettes from WT (col-0) (**A**), *shr-2* mutant (**B**) and *scr-4* mutant (**C**). Scale bar represents 1 cm. (**D**) Mean flower plastochron in WT (n=21), *shr-2* mutant (n=20) and *scr-4* mutant (n=22). (**E–G**) Heat-map quantification of the cell area in the meristem region at 5 weeks after germination from WT (n=7) (**E**), *shr-2* mutant (n=5) (**F**) and *scr-4* mutant (n=6) (**G**). Cell walls were stained with PI (gray). Scale bar represents 50 μm. (**H**) Quantification of shoot apical meristem size at 5 weeks after germination from WT (n=28), *shr-2* mutant (n=30) and *scr-4* mutant (n=30). (**I–K**) Representative 3D projection of shoot apical meristems at 5 weeks after germination expressing the three PlaCCI markers: *pCDT1a:CDT1a-eCFP* (blue), *pHTR13:pHTR13-mCherry* (red), and *pCYCB1;1:NCYCB1;1-YFP* (green) in WT (n=11) (**I**), *shr-2* mutant (n=4) (**J**) and *scr-4* mutant (n=6) (**K**). White circles in (**I**), (**J**) and (**K**) mark the meristem region. Cell walls were stained with DAPI (gray). Scale bar represents 50 μm. (**L**)

*Figure 1 continued on next page*

*Figure 1 continued*

Quantification of epidermal cell number in the meristem region of WT (n=11), *shr-2* mutant (n=4) and *scr-4* mutant (n=6). (**M–O**) Representative 3D projection of shoot apical meristems at 5 weeks after germination expressing the auxin response reporter *pDR5rev:3XVENUS-N7* in WT (Col-0) (n=6) (**M**), *shr-2* mutant (n=4) (**N**) and *scr-4* mutant (n=4) (**O**). Yellow arrowheads in (**M**), (**N**) and (**O**) show primordia with pDR5rev:3XVENUS-N7 expression. Cell walls were stained with PI (red). Scale bar represents 50 μm. (**P**) Quantification of cells in different cell cycle phases in the meristem region (area surrounded by white circles in (**I**), (**J**) and (**K**)) of WT (n=11), *shr-2* mutant (n=4) and *scr-4* mutant (n=6). (**Q–S**) Representative 3D projection of shoot apical meristems at 5 weeks after germination expressing the auxin input sensor R2D2 showing DII/mDII ratio intensity in WT (Col-0) (n=4) (**Q**), *shr-2* mutant (n=4) (**R**) and *scr-4* mutant (n=3) (**S**). Yellow arrowheads in (**Q**), (**R**) and (**S**) show primordia with low auxin. Cell walls were stained with DAPI (gray). Scale bar represents 50 μm. (**T**) Quantification of auxin maxima in WT (n=6), *shr-2* mutant (n=4) and *scr-4* mutant (n=4). Asterisks indicate a significant difference (****$p < 0.0001$: Statistically significant differences were determined by Student's *t*-test). Error bars display SD.

The online version of this article includes the following figure supplement(s) for figure 1:

**Figure supplement 1.** The *shr* and *scr* mutants phenotypes.

## Exploring expression patterns and protein complexes of SHR and SCR in shoot and flower meristems

To gain insights into the precise expression patterns of SHR and SCR in the inflorescence meristem, we made use of translational reporter lines which functionally complement the known root phenotypes of the corresponding *shr* or *scr* mutants (*Long et al., 2017*). At 5 weeks after germination, expression of *pSHR:YFP-SHR* was detected in initiating flower primordia, in floral organ primordia and in diverse flower organs (*Figure 2A*). The transcriptional reporter *pSHR:ntdTomato* (*Möller et al., 2017*) showed expression from P1 onwards in the third meristematic cell layer (L3) and, weakly, in the L2 of older primordia (*Figure 1C*; *Figure 2—figure supplement 1A*). Notably, *SHR* expression was absent in the center of the SAM, floral meristems, and stem cell domains (*Figure 2A*, left and right insets).

The translational reporter *pSHR:YFP-SHR* indicated SHR protein localization throughout L3 cells (*Figure 2A and D*), with a preference for nuclei in L2 and L1 cells, where *SHR* is normally not expressed (*Figure 2C*). This suggests that the SHR protein is mobile and moves from inner to outer cell layers of lateral organ primordia. In p*SCR:H2B-YFP* transgenic *Arabidopsis* meristems, we observed fluorescent signal primarily in differentiated vasculature and sporadically in some flower primordia (*Figure 2—figure supplement 1B*, inset). With the translational reporter line *pSCR:SCR-YFP*, which complements all *scr* mutant phenotypes (*Long et al., 2017*), we detected SCR-YFP fluorescence in the nuclei of L1 cells in the central zone of the SAM, extending into deeper meristem layers of the peripheral zone and lateral organ primordia (*Figure 2B*, inset). Notably, SCR-YFP was absent in the deeper regions of the meristem center and the rib meristem. Stronger *SCR* expression was found in the lateral organ boundary region (*Figure 2E*). Hence, SHR and SCR proteins localise in partially overlapping domains during lateral organ primordia development, with SHR absent from the SAM center, but SCR present in the L1 (compare *Figure 2A*, right inset, with *Figure 2B*, inset). SHR and SCR predominantly co-localized in the nuclei of cells in the L1, L2, and L3 layers of lateral organ primordia (*Figure 2F*; *Figure 2—figure supplement 2A*). To assay for protein interactions, we conducted FRET-FLIM studies in *Arabidopsis* shoot meristems using YFP-SHR as the fluorescent donor and SCR-RFP as the acceptor. We found that YFP-SHR and SCR-RFP interact in the lateral organ primordia, petal primordia and sepal primordia (*Figure 2H*). In contrast, negative controls with non-interacting proteins (*Long et al., 2017*), SAMs from plants coexpressing *pSCR:SCR-YFP* with *pSCR:SCR-RFP* (*Figure 2G*), did not exhibit significant changes in fluorescence lifetime (*Figure 2I*). We then investigated the regulatory relationship between SHR and SCR. Comparing SCR protein localization using the *pSCR:SCR-YFP* reporter line in WT and *shr-2* mutant backgrounds, we observed that SHR depletion in the central zone of the SAM did not affect SCR protein levels (*Figure 2—figure supplement 2D, E and L*). However, in lateral organ primordia, SCR protein expression was significantly reduced in *shr-2* mutants compared to WT (*Figure 2—figure supplement 2F, G and M*). qRT-PCR analysis of SCR transcript levels in SAMs further supported the significant decrease in *shr-2* mutants (*Figure 2—figure supplement 2N*). These findings indicate that SHR acts as a transcriptional activator of SCR in the SAM, while SCR expression in the meristem center is independent of SHR. Interestingly, the expression patterns of *pSHR:ntd-TOMATO* and *pSCR:SCR-YFP* do not completely overlap, suggesting that SHR functions as a mobile

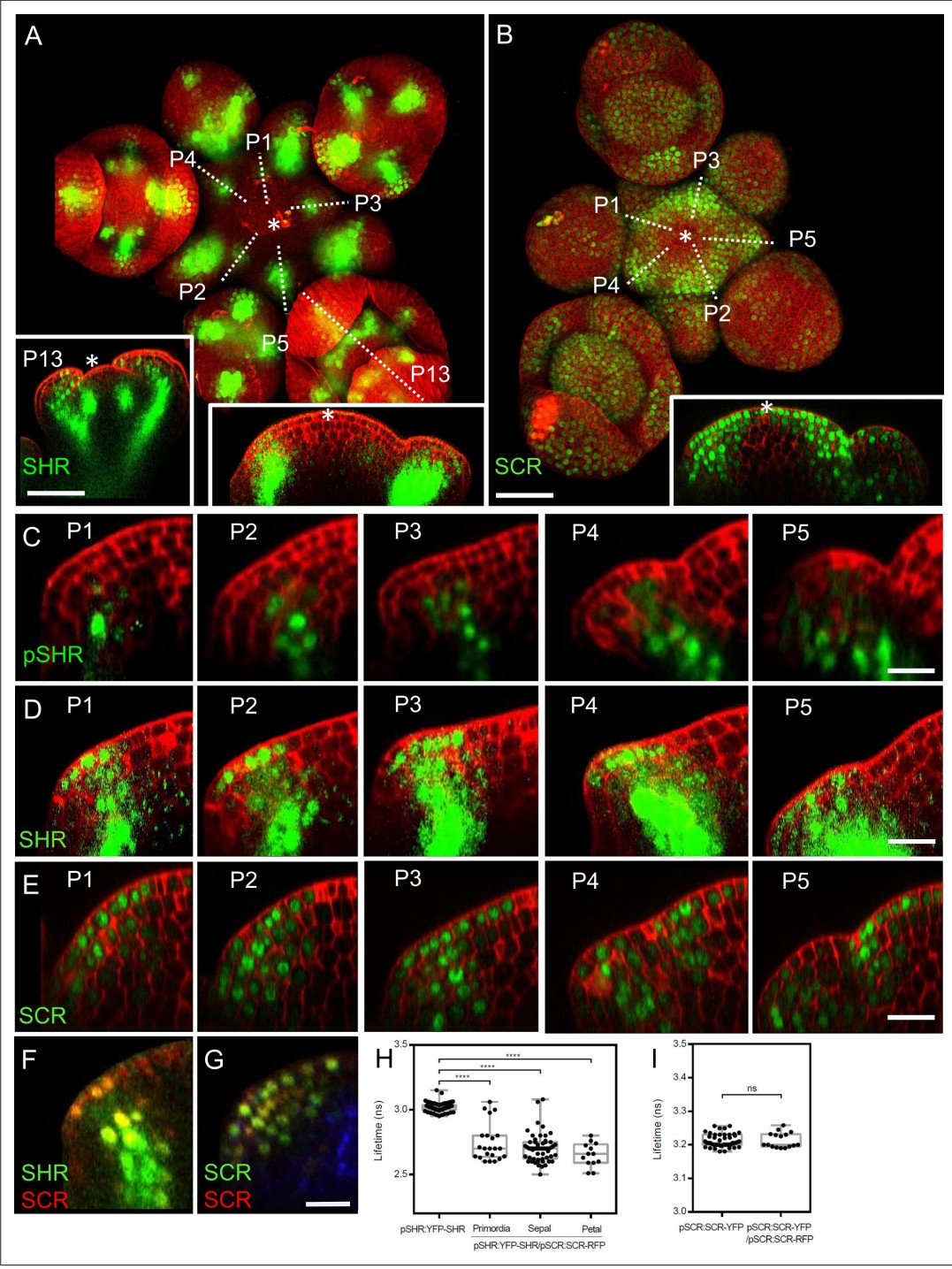

**Figure 2.** Expression patterns of SHR and SCR in the shoot apical meristem and In vivo FRET–FLIM quantification of SHR–SCR association in the shoot apical meristem. (**A**) Representative 3D projection of shoot apical meristem at 5 weeks after germination expressing *pSHR:YFP-SHR* reporter (green) (n≥6). The lower right inset shows a longitudinal optical section through the middle of the SAM. The lower left inset shows a longitudinal optical section through the middle of primordia 13 (representative section orientation shown by dotted line). Cell walls were stained with DAPI (red). Scale bar represents 50 µm. (**B**) Representative 3D projection of shoot apical meristem at 5 weeks after germination expressing *pSCR:SCR-YFP* reporter (green) (n≥10). The lower right inset shows longitudinal optical section through the middle of the SAM. Cell walls were stained with PI (red). Scale bar represents 50 µm. (**C**) Longitudinal optical sections through the middle of five successive primordia expressing *pSHR:ntdTomato* reporter (green) (representative section orientation shown by dotted line in *Figure 2—figure*

*Figure 2 continued on next page*

*Figure 2 continued*

*supplement 1A*). Cell walls were stained with DAPI (red). Scale bar represents 20 μm. (**D** and **E**) Longitudinal optical sections through the middle of five successive primordia expressing *pSHR:YFP-SHR* reporter (green) (**C**) and *pSCR:SCR-YFP* reporter (green) (**D**) (representative section orientation shown by dotted line in (**A**) and (**B**), respectively). Scale bars represent 20 μm. *P*=Primordium. (**F** and **G**) Longitudinal optical sections through the middle of five successive primordia coexpressing *pSHR:YFP-SHR* reporter (green) and *pSCR:SCR-RFP* reporter (red) (**F**), and *pSCR::SCR:YFP* reporter (green) and *pSCR::SCR:RFP* reporter (red) (**G**). Chlorophyll (blue). Scale bar represents 20 μm. (**H**) Average lifetime of YFP-SHR when expressed alone (pSHR:YFP-SHR (n =75)), or coexpressed together with SCR-RFP (pSHR:YFP-SHR/pSCR:SCR-RFP) in lateral organ primordia (n=22), sepal primordia (n=55) and petal primordia (n=13) in the shoot meristem. (**I**) Average lifetime of SCR-YFP when expressed alone (pSCR:SCR-YFP), or coexpressed together with SCR-RFP (pSCR:SCR-YFP/pSCR:SCR-RFP) in shoot meristem. Asterisks indicate a significant difference (****p<0.0001: Statistically significant differences were determined by Student's *t*-test, ns = no significant difference). *P*=Primordium.

The online version of this article includes the following figure supplement(s) for figure 2:

**Figure supplement 1.** The expression pattern of SHR and SCR in the shoot apical meristem.

**Figure supplement 2.** SHR regulates *SCR* expression in the shoot apical meristem.

protein in the SAM (*Figure 2—figure supplement 2C*). Moreover, we investigated the influence of SCR on SHR protein movement by examining SAMs coexpressing *pSHR:SHR-YFP* and *pSCR:SCR-RFP*. Our analysis revealed that SHR-YFP was enriched in the nuclei of cells coexpressing SCR-RFP in lateral organ primordia (*Figure 2—figure supplement 2C and K*). This enrichment suggests that SCR may trap SHR in the nuclei, thereby limiting its intercellular movement.

## Interplay between JKD and the SHR-SCR complex in the SAM

We investigated the roles of the BIRD-family transcription factor JACKDAW (JKD) in SAM development. SAMs of *jkd-4* mutants were larger than WT, due to an increased cell number (*Figure 3B–E*, and *Figure 3—figure supplement 1A*). The transcriptional reporter *pJKD:YFP-RFP* and the translational *pJKD:JKD-YFP* reporter, previously shown to complement the *jkd-4* mutant phenotype (*Long et al., 2017*) showed that *JKD* was expressed in some cells of the peripheral zone and in the abaxial domain of flower primordia (*Figure 3A*). Expression patterns of transcriptional and translational reporter lines were mostly identical (*Figure 3A*; *Figure 3—figure supplement 1E*), indicating that *JKD* RNA and protein are found in the same cells and that JKD protein is not mobile. Notably, the JKD expression domain expanded from a few cells at the abaxial side of P1 into a ring-like abaxial domain in sepals and, at later stages, in stamen and carpel primordia (*Figure 3—figure supplement 1E–G*).

We next investigated if early *JKD* expression correlates with sepal identity. To address this, we used the strong *lfy-12* mutant, which generates mostly sepal-like organs in the flowers (*Weigel et al., 1992*), and the *clv3-9* mutant, characterized by an excessively enlarged SAM and flowers with additional organs (*Schlegel et al., 2021*). *JKD* was confined to sepal primordia in *clv3-9* mutants and to the sepal-like organs in *lfy-12* mutants, similar to the WT (*Figure 3—figure supplement 1B–D*). We then explored potential molecular interactions of JKD with SCR in the SAM via FRET-FLIM experiments in vivo. SAMs coexpressing *pJKD:JKD-YFP* and *pSCR:SCR-RFP* (*Figure 3—figure supplement 1*) showed significant fluorescence lifetime reductions of up to 0.17±0.02 ns, compared to the JKD-YFP donor-only (*Figure 3N*). This indicates that the SHR-SCR complex can form complexes with JKD in the SAM periphery, floral organ primordia, and floral organs. We propose that JKD may modulate or even dampen the activities of the SHR-SCR complexes, since JKD and SHR-SCR exert opposite effects on meristem size. To better understand if SHR-SCR-JKD complexes affect cell division rates during lateral organ primordia development, we studied the expression of CYCLIN genes as key regulators of cell division (*pCYCD1;1:GFP, pCYCD2;1:GFP, pCYCD3;2:GFP, pCYCD3;3:GFP, pCYCD5;1:GFP, pCYCD6;1:GFP, pCYCD7;1:GFP* and *pCYCB1;1:CYCB1;1-GFP*) (*Dewitte et al., 2003*; *Ubeda-Tomás et al., 2009*). Only *CYCD1;1, CYCD3;2, CYCD3;3, CYCD6;1* and *CYCB1;1* were visibly expressed in the SAM (*Figure 3—figure supplement 2A–D and G*). *CYCD1;1, CYCD3;2* and *CYCB1;1* exhibited a patchy expression pattern in sepal primordia (*Figure 3—figure supplement 2A, B and D*), while *CYCD3;3* was more uniformly distributed with an enrichment in lateral organ primordia (*Figure 3—figure supplement 2C*). *pCYCD6;1:GFP* expression was detected in the L3 layer of lateral organ primordia and in sepal primordia from all stages, but not at the center of the SAM (*Figure 3F, F' and F"*).

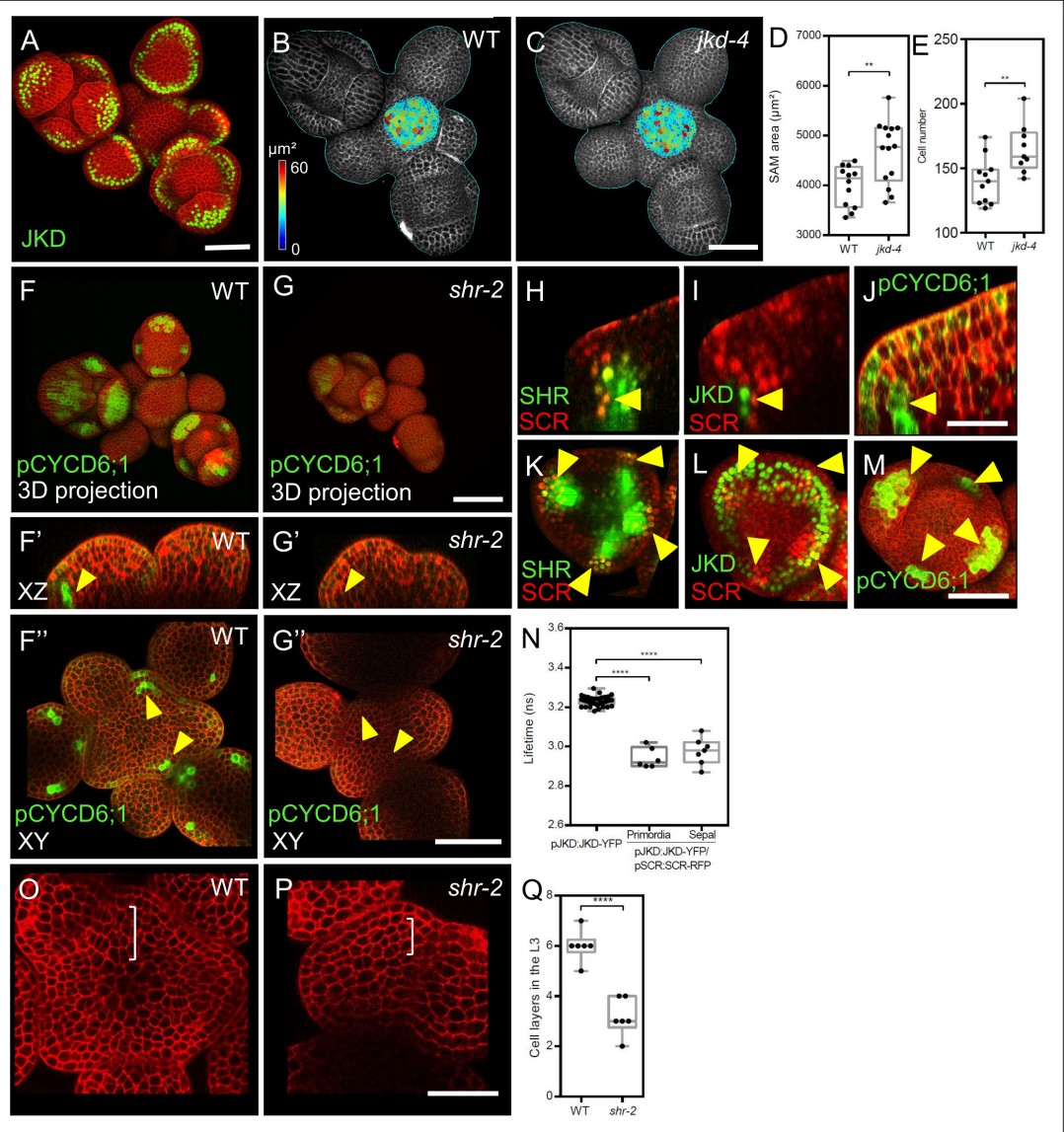

**Figure 3.** JKD functions and expression pattern in the shoot apical meristem. SHR regulates *CYCD;1* expression in the shoot apical meristem. (**A**) Representative 3D projection of shoot apical meristems at 5 weeks after germination expressing *pJKD:JKD-YFP* reporter (green) (n≥20). Cell walls were stained with PI (red). Scale bar represents 50 μm. (**B** and **C**) Heat-map quantification of the cell area in the meristem region at 5 weeks after germination from WT (**B**) (n=10) and *jkd-4* mutant (**C**) (n=10). Cell walls were stained with PI (gray). Scale bar represents 50 μm. (**D**) Quantification of shoot apical meristem size at 5 weeks after germination from WT (n=12) and *jkd-4* mutant (n=14). (**E**) Quantification of epidermal cell number in the meristem region of WT (**B**) (n=10) and *jkd-4* mutant (**C**) (n=10). (**F** and **G**) Representative 3D projection of shoot apical meristems at 5 weeks after germination expressing *pCYCD6;1:GFP* reporter (green) in WT (n=5) (**F**) and *shr-2* mutant (n=6) (**G**). cell walls were stained with PI (red). Scale bar represents 50 μm. (**F'** and **G'**) Longitudinal optical sections of (**F**) and (**G**) respectively. (**F''** and **G''**) Transversal optical sections of (**F**) and (**G**) respectively. Scale bar represents 50 μm. (**H–J**) lateral organ primordia showing coexpression of *pSHR:SHR-YFP* reporter (green) and *pSCR::SCR-RFP* reporter (red) (**H**), *pJKD:JKD-YFP* reporter (green) and *pSCR::SCR-RFP* reporter (red) (**I**) and *pCYCD6;1-GFP* reporter (green) (**J**). Scale bar represents 20 μm. (**K–M**) Florescence meristem stage 4 of flower development showing coexpression of *pSHR:SHR-YFP* reporter (green) and *pSCR::SCR-RFP* reporter (red) (**K**), *pJKD:JKD-YFP* reporter (green) and *pSCR::SCR-RFP* reporter (red) (**L**) and *pCYCD6;1-GFP* reporter (green) (**M**). Scale bar represents 20 μm. (**N**) Average lifetime of JKD-YFP when expressed alone (pJKD:JKD-YFP (n = 38)), or coexpressed together with SCR-RFP (pJKD:JKD-YFP/pSCR:SCR-RFP) in lateral organ primordia (n=6) and sepal primordia (n=7) in the shoot meristem. (**O** and **P**) Transversal optical sections of the inflorescence apex at 5 weeks after germination from WT (**O**) and

*Figure 3 continued on next page*

*Figure 3 continued*

*shr-2* mutant (**P**). Scale bar represents 20 µm. (**Q**) Quantitative comparison of cell files within the L3 in optical sections of the inflorescence apex from WT (n=6) and *shr-2* mutant (n=6). Asterisks indicate a significant difference (****p<0.0001; **p<0.001: Statistically significant differences were determined by Student's *t*-test,).

The online version of this article includes the following figure supplement(s) for figure 3:

**Figure supplement 1.** The expression pattern of JKD in *lfy* and *clv3* mutant shoot apical meristems and colocalization of the expression of JKD and SCR in the shoot apical meristem.

**Figure supplement 2.** The colocalization of the expression patterns of SHR, SCR, JKD and CYCD6;1 in the shoot apical meristem.

Importantly, *pCYCD6;1:GFP* expression overlapped with the expression domains of *SHR*, *SCR*, and *JKD* (**Figure 3H–M**; **Figure 3—figure supplement 2H–S**). In *shr-2* mutants, we found reduced *pCYC-D6;1:GFP* expression in young sepals, but no expression in lateral organ primordia, (**Figure 3F–F"** *and G–G"*).

Importantly, we found fewer cell layers in the L3 of *shr-2* mutant lateral organ primordia compared to WT (**Figure 3O–Q**). This parallels our findings in the *cycd6;1* mutant, where we observed a diminished meristem size and a reduced number of cell layers in the L3 compared to the WT (**Figure 4—figure supplement 2I–N**). These findings align with the presumed function of SHR-SCR-JKD complexes in regulating cell division within the SAM.

## Auxin induces SHR and SCR expression in the SAM

Since we observed that *SHR* and *SCR* controls SAM size and lateral organ primordia initiation, but also auxin accumulation, which is triggered by local auxin accumulation, we investigated how SHR-SCR interacts with auxin distribution and/or signaling. PIN1 is an auxin efflux transporter which controls the flow and the distribution of auxin across the SAM (**Benková et al., 2003**). Expression patterns of *SHR, SCR* and PIN1 overlapped in primordia (**Figure 2A–E**; **Figure 4—figure supplement 1A–F**), indicating that the *SHR* and *SCR* expression patterns may be guided by auxin signaling. In silico analysis of the *SHR* promoter sequence identified two putative auxin-response element (AuxRE) core motifs (TGTCTC) (**Figure 4H**), which could recruit ARF transcription factors for auxin dependent expression (**Schlereth et al., 2010**; **Zhao et al., 2010**; **Ulmasov et al., 1999**). Treatment of WT inflorescences with 10 µM of the synthetic auxin indole-3-acetic acid (IAA) caused expression of *SHR* and *SCR to increase* within 12 hr. We then analyzed SAMs expressing *pSHR:SHR-YFP* and *pSCR:SCR-YFP* after treatment with 10 µM of the synthetic auxin analogue 2,4D, an auxin analogue that can freely enter and exit cells without the need for transporter. We found that expression visibly increased within 2–5 hr after the start of the hormone treatment (**Figure 4B–G**), but the expression patterns of both *SHR* and *SCR* were not strongly altered. Since we observed an overlap of auxin accumulation domains with those of *SHR* and *SCR*, and also established that auxin induces *SHR* and *SCR* expression, we tested if changes in auxin transport and local distribution in turn affected *SHR* and *SCR* expression patterns. We inhibited polar auxin transport and hence its distribution in tissues using N-1-naphthylphthalamic acid (NPA). The mock control SAMs expressing *pPIN1:PIN1-GFP* showed a wide distribution of *PIN1-GFP*, which relocated into a ring shaped domain in deeper regions of the meristem 3d after treatment with 100 µM NPA (**Figure 4—figure supplement 2E–F"**). Concomitantly, the expression pattern of *SHR-YFP* changed from being associated with primordia into a ring-shaped domain, reflecting the rearrangement of the *PIN1* expression domain (**Figure 4—figure supplement 2A–B"**). Similar changes in expression pattern were found for *SCR* upon NPA treatment (**Figure 4—figure supplement 2C–D"**). NPA treatment resulted in enlarged expression domains for *pCYCD6;1:GFP*, a key target gene of the SHR-SCR-JKD complex (**Figure 4—figure supplement 2G–H"**). Together, these findings suggest that auxin distribution and signaling coordinate levels and expression patterns of SHR and SCR in the SAM.

## MONOPTEROS regulates SHR and SCR expression in the SAM

The expression patterns of *MP*, *SHR*, and *SCR* partially overlapped in lateral organ primordia (**Figure 2D and E**; **Figure 4—figure supplement 1A, B, D and E**), and we therefore analyzed the expression of *pSHR:ntdTomato* in *mp-B4149*, a loss-of-function mutant for *MP*. In WT vegetative meristems at 7 DAG, *SHR* was expressed in the deeper region of young leaf primordia, where the

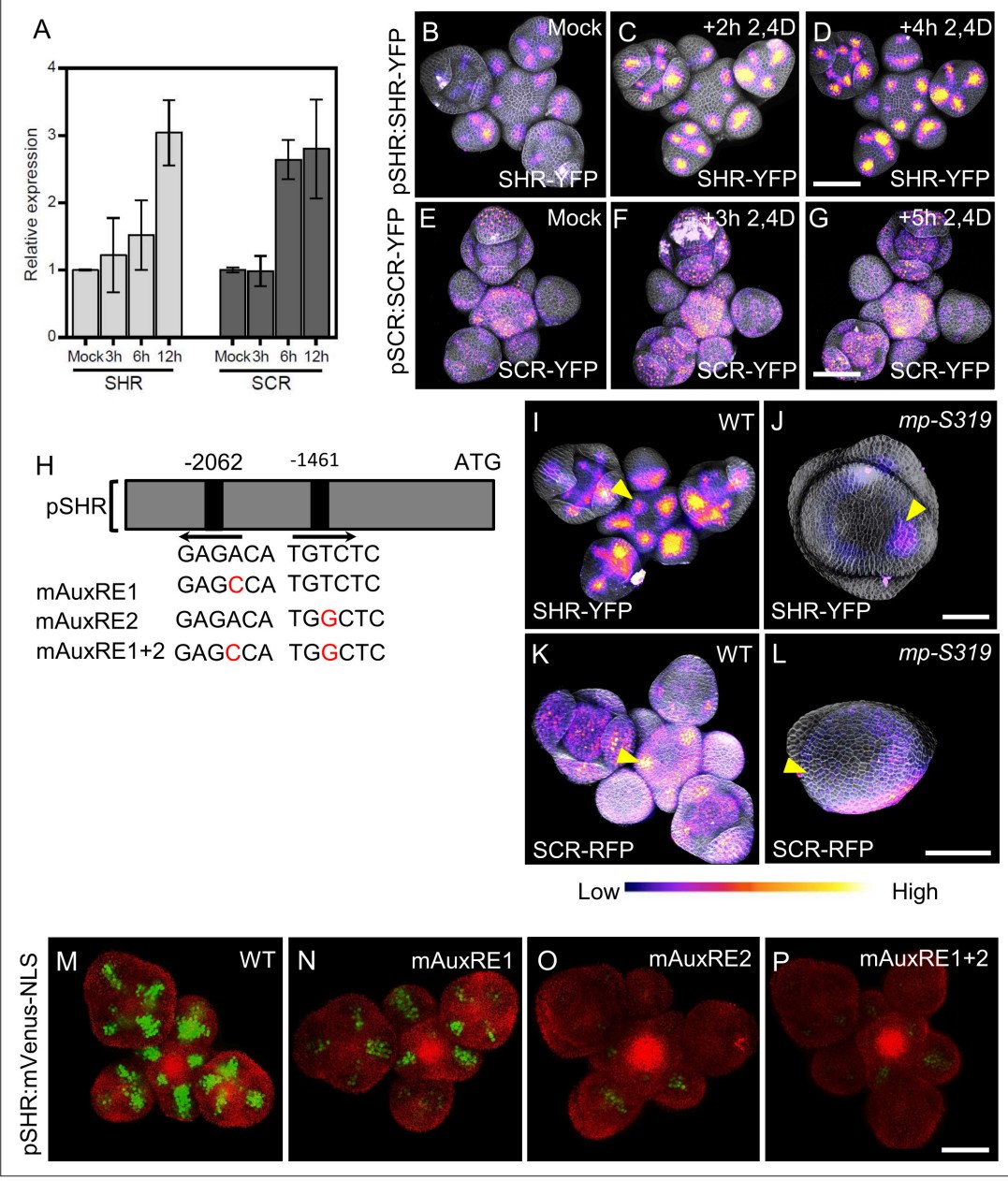

**Figure 4.** MP Regulates *SHR* and *SCR* expression in the shoot apical meristem. (**A**) Quantitative real-time PCR analysis showing the relative expression levels of *SHR* and *SCR* expression in response to auxin (10 µm IAA) in WT shoot apical meristems. The expression level in Col-0 is set to 1 and error bars show standard deviation. Expression levels were normalized using AT4G34270 and AT2G28390. (**B–D**) Representative 3D projection of shoot apical meristems at 5 weeks after germination expressing *pSHR:SHR-YFP* reporter (magenta) in mock (**B**) (n=4), after 2 hours 10 µM 2,4D treatment (n=3) (**C**) and after 4 hours 10 µM 2,4D treatment (n=3) (**D**). Fluorescence intensities were coded blue to yellow corresponding to increasing intensity. Cell walls were stained with PI (gray). Scale bar represents 50 µm. (**E–G**) Representative 3D projection of shoot apical meristems at 5 weeks after germination expressing *pSCR:SCR-YFP* reporter (magenta) mock (**E**) (n=4), after 3 hours 10 µM 2,4D treatment (n=2) (**F**) and after 5 hours 10 µM 2,4D treatment (n=2) (**G**). Fluorescence intensities were coded blue to yellow corresponding to increasing intensity. Cell walls were stained with PI (gray). Scale bar represents 50 µm. (**H**) Schematic representation of the *SHR* promoter. The positions of two auxin response elements are shown. Overview of mutated promoter versions of *pSHR*. AuxREs were mutated and multiple combinations of these mutated motifs were combined into a single promoter. The original AuxRE sequence GAGACA was mutated to GAGCCA (mAuxRE1), the original AuxRE sequence TGTCTC was mutated to TGGCTC (mAuxRE2). (**I and J**) Representative 3D projection of shoot apical meristems at 5 weeks after germination expressing *pSHR:SHR-YFP* reporter (magenta) in WT (n=4) (**I**) and *mp-S319*

*Figure 4 continued on next page*

*Figure 4 continued*

mutant (n=3) (**J**). Yellow arrowheads in (**I**) and (**J**) indicate the region where flower primordia initiate and pSHR:SHR-YFP expression. Fluorescence intensities were coded blue to yellow corresponding to increasing intensity. Cell walls were stained with PI (gray). Scale bar represents 50 µm. (**K** and **L**) Representative 3D projection of shoot apical meristems at 5 weeks after germination expressing *pSCR:SCR-RFP* reporter (magenta) in WT (n=4) (**K**) and *mp-S319* mutant (n=4) (**L**). Yellow arrowheads in (**K**) and (**L**) indicate the region where flower primordia initiate and pSCR:SCR-RFP expression. Fluorescence intensities were coded blue to yellow corresponding to increasing intensity. Cell walls were stained with DAPI (gray). Scale bar represents 50 µm. (**M–P**) Representative 3D projection of shoot apical meristems at 5 weeks after germination expressing mVenus-NLS under the control of the wild-type SHR promoter (n≥6) (**M**), and under the control of the SHR promoter with mutations in AuxRE motifs mAuxRE1 (n≥5) (**N**) mAuxRE2 (**O**) and mAuxRE1 + 2 (n≥5) (**P**); Chlorophyll (red). Scale bar represents 50 µm.

The online version of this article includes the following figure supplement(s) for figure 4:

**Figure supplement 1.** The expression patterns of SHR, SCR, MP and PIN1 in the shoot apical meristem.

**Figure supplement 2.** *CYCD6;1* expression responds to auxin.

**Figure supplement 3.** MP Regulates *SHR* and *SCR* expression in the shoot apical meristem.

**Figure supplement 4.** *shr* mutant and *shr mp-S319* double-mutant phenotypes.

**Figure supplement 5.** MP induces the expression of SHR in planta.

**Figure supplement 6.** MP Regulates SHR expression in the shoot apical meristem.

**Figure supplement 7.** MP induces the expression of SHR in planta.

**Figure supplement 8.** The expression pattern of the different promoter versions of SHR in the root apical meristem.

**Figure supplement 9.** LFY act downstream of SHR in the shoot apical meristem.

vasculature will eventually develop. *SHR* was expressed in a similar pattern in the *mp-B4149* mutants, albeit at lower levels (*Figure 4—figure supplement 3A and B*), indicating that *MP* is required for normal *SHR* expression levels post embryogenesis. Because *mp-B4149* mutants fail to develop beyond the early seedling stage, we used the hypomorphic allele *mp-S319* to study later stages of development. *mp-S319* mutants display weaker phenotypes than *mp-B4149* null mutants and initiate an inflorescence meristem with occasionally some flower primordia. We used the transcriptional reporter *pSHR:ntdTOMATO* and the translational reporter *pSHR:SHR-YFP* to distinguish between effects of auxin on *SHR* promoter activity and post-transcriptional effects leading to altered protein localization. Compared to WT, *pSHR:ntdTOMATO* expression was strongly reduced in *mp-S319* inflorescences (*Figure 4—figure supplement 3C–D'*). Similar results were observed for the translational reporter *pSHR:SHR-YFP*, and weak expression in a ring-shaped domain was found in cross sections through the inflorescence stem (*Figure 4I–J*; *Figure 4—figure supplement 3E and F*). Similarly, using *pSCR:SCR-RFP*, we observed reduced expression of *SCR* in the meristem periphery in *mp-S319*, which now overall resembled the *SHR* expression pattern (*Figure 4K–L*; *Figure 4—figure supplement 3G and H*). We therefore conclude that *MP* is required for maintaining normal expression of *SHR* and *SCR* expression in the meristem periphery and primordia. We then tested whether *MP* could promote *SHR* expression throughout the meristem, using an inducible MP-GR fusion protein expressed from the *Ubiquitin10* promoter (*pUBQ10:MP-GR*). Treatment of these plants with dexamethasone (DEX) allows the nuclear entry of MP-GR. Within 4 hr of DEX treatment, ubiquitously expressed MP resulted in increased expression levels of *pSHR:SHR-YFP* and *pSHR:ntdTOMATO* while SHR retained its WT expression domain (*Figure 4—figure supplement 3I–L*). Thus, MP exerts a quantitative regulatory control over *SHR* expression. To better understand the distinct roles of *SHR* and *MP* in organ initiation, we generated double mutants of *mp-S319* and *shr-2*. Unlike *mp-S319* that forms only fewer flower primordia than WT, the phenotype of the double mutant *mp-S319; shr-2* was enhanced with naked inflorescences that lacked all organ primordia (*Figure 4—figure supplement 4A–D''*). To test if MP can directly activate *SHR* expression, we tested if the two candidate AuxREs (AuxRE1 and AuxRE2) in the *SHR* promoter can control expression of the reporter genes *mVenus* or *Luciferase* upon Agrobacterium-mediated transient expression in *Nicotiana benthamiana* leaves. *pSHR(WT):mVenus* was co-infiltrated with *p35S:MP*, or with *p35S:GUS* as a negative control. We observed strong *mVenus* expression from the WT-*SHR* promoter in the presence of *p35S:MP*, which was about sixfold higher than upon coexpression of *p35S:GUS* (*Figure 4—figure supplement 5B*). Mutating AuxRE1

from 'GAGACA' to 'GAGCCA' (mAuxRE1) or 'GTGCTC' (mAuxRE1-2) and AuxRE2 from 'TGTCTC' to 'TGGCTC' (mAuxRE2) or 'TGGAGA' (mAuxRE2-2) by site directed mutagenesis drastically reduced the response to *p35S:MP* coexpression, and mutating both AuxREs had an additive effect, resulting in no significant response to overexpression of *MP* (*Figure 4—figure supplement 5A–J* and *Figure 4—figure supplement 6A-J*). Similar results were obtained using a *Luciferase* assay system (*Figure 4—figure supplement 7A-H*), indicating that MP can interact with the two AuxRE elements in the *SHR* promoter. To investigate if these elements also contribute to *SHR* expression in *Arabidopsis*, we generated transgenic plants using either the WT *SHR* promoter, or the mutant versions carrying base changes in one or both AuxREs. At least 10 independent transgenic lines were analyzed for each promoter construct. Expression of the WT *pSHR:mVenus-NLS* produced a strong expression in lateral organ primordia and floral meristems (*Figure 4M*; *Figure 4—figure supplement 6K*). However, mutations in AuxRE1 (mAuxRE1 or mAuxRE1-2) or AuxRE2 (mAuxRE2 or mAuxRE2-2) led to decreased *mVenus* expression in all organ primordia stages and the floral meristem (*Figure 4N and O*; *Figure 4—figure supplement 6L and M*). When both AuxREs (mAuxRE1 +2 or mAuxRE1−2+2–2) were mutated, *mVenus* expression was barely detectable (*Figure 4P*; *Figure 4—figure supplement 6N*). We concluded that both AuxRE motifs function in an additive manner during activation of *SHR* expression in the SAM. Notably, consistent with the expression in the SAM, mutations in AuxREs motifs in the *SHR* promoter also led to a decreased *mVenus* expression in the RAM compared to the WT *pSHR:mVenus-NLS*, where *SHR* and *MP* expression overlap ( *Figure 4—figure supplement 8A-H*).

## *SHR* regulates *LFY* expression in lateral organ primordia

Floral identity of lateral organ primordia depends on the TF LEAFY (LFY), which is directly activated by MP (*Yamaguchi et al., 2013*). Since MP induces also *SHR* expression, we asked how *SHR* and *LFY* activities are coordinated. In WT, *SHR* and *SCR* were expressed two plastochrons prior to *LFY* in the primordia (*Figure 4—figure supplement 9A*), and we hypothesized that *SHR* might mediate MP-dependent *LFY* expression. We first tested whether *LFY* expression was altered in *shr-2* mutants. mRNA quantification via qRT-PCR from SAM and flower primordia up to stage 5 revealed that *LFY* RNA levels were downregulated by approximately 60% in *shr-2* compared to WT (*Figure 4—figure supplement 9D*). We further tested the LFY protein level and expression pattern using *pLFY:GFP-LFY* reporter (previously named *pLFY::GLFY*) (*Wu et al., 2003*). In WT, *LFY* was not expressed in the SAM, but in flower primordia from stage 3 onwards. At later stages, *LFY* remained expressed in the adaxial side of sepals, and in the 2nd and 3rd floral whorls. In *shr-2* mutants, *GFP-LFY* fluorescence was barely visible at early stages and strongly decreased at later stages, compared to WTand (*Figure 4—figure supplement 9B and C*). This indicates that *SHR* function is required for normal *LFY* expression patterns. *SHR* and *SCR* expression was unaltered in *lfy-12* mutants (*Figure 4—figure supplement 9M-P*), and we therefore conclude that *LFY* does not feed-back to the *SHR-SCR* network. In the *lfy-12* null-mutant, leaf-like bracts are generated from the inflorescence meristem and formation of flower meristems is delayed. Most floral organs are leaf-like and not arranged in regular whorls. Double mutants of *lfy-12; shr-2* were strongly retarded in growth and displayed dramatically enhanced floral primordium initiation defects (*Figure 4—figure supplement 9E-L*). We conclude from this analysis that the *SHR-SCR* network promotes *LFY* expression, and that it also promotes floral meristem development in a *LFY*-independent manner.

## *SCL23* interacts with the *SHR-SCR* pathway and contributes to SAM size maintenance

The closest homologue of SCR is the GRAS-family TF SCARECROW-LIKE23 (SCL23). We investigated the expression pattern of *SCL23* in the SAM using the transcriptional reporter *pSCL23:H2B-YFP*, and the functional translational reporter line *pSCL23:SCL23-YFP* (*Long et al., 2015a*). *pSCL23:H2B-YFP* was expressed in the L3 of the SAM and floral meristems, and in floral organ primordia (*Figure 5A and A'*). The translational reporter *pSCL23:SCL23-YFP* showed a wider expression pattern than the transcriptional reporter (*Figure 5B and B'*), suggestive of SCL23 protein mobility. To test if the wider expression pattern of the translational reporter line is due to the presence of additional transcriptional control regions within the coding sequence of *SCL23*, we expressed *SCL23-mVenus* under the *CLV3* promoter, which confines the expression exclusively to the central zone. We observed SCL23 spreading from the expression domain in the central zone into the surrounding cells, consistent with

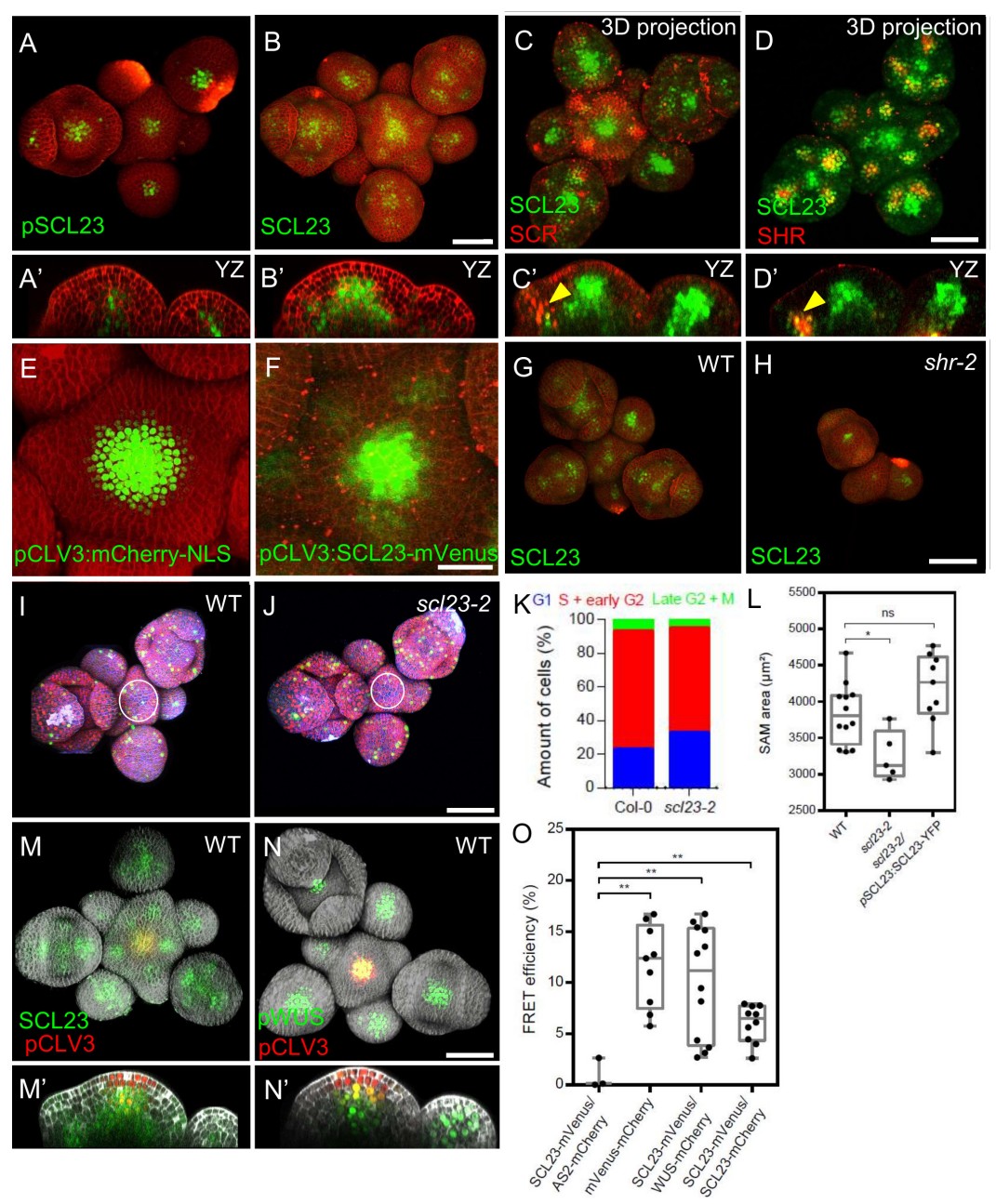

**Figure 5.** Interplay of SCL23, SCR, SHR, SCL23-WUS Interaction in the Shoot Apical Meristem. (**A** and **B**) Representative 3D projection of shoot apical meristems at 5 weeks after germination expressing *pSCL23:H2B-YFP* reporter (green) (n≥3) (**A**) and *pSCL23:SCL23-YFP* reporter (green) (n≥10) (**B**). Cell walls were stained with PI (red). Scale bar represents 50 µm. (**A'** and **B'**) Longitudinal optical sections of (**A**) and (**B**) respectively. (**C** and **D**) Representative 3D projection of shoot apical meristems at 5 weeks after germination meristems coexpressing *pSCL23:SCL23-YFP* reporter (green) and *pSCR:SCR-RFP* reporter (red) (n≥3) (**C**) and *pSCL23:SCL23-YFP* reporter (green) and *pSHR:mScarlet-RFP* reporter (red) (n≥3) (**D**). Cell walls were stained with PI (red). Scale bar represents 50 µm. (**C'** and **D'**) Longitudinal optical sections of (**C**) and (**D**) respectively. (**E** and **F**) Representative 3D projection of shoot apical meristems at 5 weeks after germination expressing *pCLV3-mCherry-NLS* reporter (green) (n≥5) (**E**) and the *pCLV3:SCL23-mVenus* reporter (green) (n≥4) (**F**). Cell walls were stained with PI (red). Scale bar represents 50 µm. (**G** and **H**) Representative 3D projection of shoot apical meristems at 5 weeks after germination expressing *pSCL23:SCL23-YFP* reporter (green) in WT (n=4) (**G**) and *shr-2* mutant (n=3) (**H**). Cell walls were stained with PI (red). Scale bar represents 50 µm. (**I** and **J**) Representative 3D projection of shoot apical meristems at 5 weeks after germination from WT (n=11) (**I**) and *scl23-2* (n=5) (**J**) coexpressing the three PlaCCI markers. *pCDT1a:CDT1a-eCFP*

*Figure 5 continued on next page*

*Figure 5 continued*

reporter (blue), *pHTR13:pHTR13-mCherry* reporter (red) and *pCYCB1;1:NCYCB1;1-YFP* reporter (green). Cell walls were stained with DAPI (gray). Scale bar represents 50 µm. (**K**) Quantification of different cell cycle phases of SAM expressing PlaCCI in WT (n=11) and *scl23-2* (n=5). Asterisks indicate a significant difference (*$P<0.01$: Statistically significant differences were determined by Student's *t*-test, ns = no significant difference). (**L**) Quantification of shoot apical meristem size at 5 weeks after germination from Col-0 (n=12), *scl23-2* mutant (n=5) and *scl23-2/ p*SCL23:SCL23-YFP (n=9). (**M** and **N**) Representative 3D projection of shoot apical meristems at 5 weeks after germination coexpressing *pSCL23:SCL23-YFP* (green) and *pCLV3-mCherry-NLS* reporter (red) (n≥3) (**M**) and *pWUS:3xVenus-NLS* reporter (green) and *pCLV3-mCherry-NLS* reporter (red) (n≥6) (**N**). Cell walls were stained with DAPI (gray). Scale bar represents 50 µm. (**M'** and **N'**) Longitudinal optical sections of (**M**) and (**N**) respectively. (**O**) FRET efficiency measured in epidermis cells of *N. benthamiana* between SCL23-mVenus and WUS-mCherry (n=12) or SCL23-mCherry (n=10), compared with the negative control SCL23-mVenus and AS2-mCherry (n=3) and positive control mVenus-mCherry (n=9). Asterisks indicate a significant difference (**$p<0.001$; *$p<0.01$: Statistically significant differences were determined by Student's *t*-test, ns = no significant difference).

The online version of this article includes the following figure supplement(s) for figure 5:

**Figure supplement 1.** SCL23 and WUS are negatively regulated by the CLV pathway in the shoot apical meristem.

**Figure supplement 2.** Mutant combinations of *shr*, *scr,* and *scl23*.

**Figure supplement 3.** Overexpression of SHR, SCR, and SCL23 in the SAM.

**Figure supplement 4.** WUS and SCL23 cooperatively control shoot stem cell homeostasis in the shoot apical meristem.

high SCL23 protein mobility in the SAM (*Figure 5E and F*). We then analyzed co-expression of SCL23 and SCR to investigate the regulatory dynamics among *SCL23*, *SCR* and *SHR*. A dual reporter line with *pSCR:SCR-RFP* and *pSCL23:SCL23-YFP* revealed that the expression patterns of both genes are in most cases mutually exclusive, with SCL23 in a central domain of the SAM and in the inner cell layers of primordia, while SCR is confined to the peripheral zone, the L1 of the central zone and the outer cell layers of lateral organ primordia (*Figure 5C and C'*). Some cells expressing both SCL23 and SCR were located deep inside lateral organ primordia (*Figure 5C'*, yellow arrowheads). A similar coexpression analysis of *pSCL23:SCL23-YFP* with *pSHR:mScarlet-SHR* indicated an overlap of expression in the deep cell layers of lateral organ primordia (*Figure 5D and D'*, yellow arrowheads). In *shr-2* mutants, *pSCL23:SCL23-YFP* remains only weakly expressed in the meristem center (*Figure 5G and H*), indicating that SHR is required to promote expression of *SCL23* in the SAM. We then performed genetic interaction studies to understand the relationship between *SCL23*, *SCR* and *SHR* in the SAM. In *scl23-2* mutants, SAM size is significantly decreased compared to WT or to the complemented line *pSCL23:SCL23-YFP/scl23-2* (*Figure 5L*; *Figure 5—figure supplement 1A–C*). Using the PlaCCI marker to analyze cell division patterns, we found that similar to *shr-2* and *scr-3* or *scr-4* mutants, the percentage of cells in G1 phase increased, indicating an extended G1 phase and delayed cell divisions in the SAM (*Figure 5I–K*). Genetic analyses showed that *shr* mutants are epistatic to *scl23* mutants in double mutant combinations, and that *scr-3; scl23-2* double mutants and *shr-2; scr-3; scl23-2* triple mutants were additive with smaller plant rosettes and reduced plant stature (*Figure 5—figure supplement 2A–H'*; *Yoon et al., 2016*). To further analyse *SCL23* function, we generated transgenic *Arabidopsis* plants overexpressing *SCL23* from the *pUBIQ10* (*pUBIQ10:SCL23-mVenus*) or *pRPS5A* (*pRPS5A:SCL23-mVenus*) promoter. Both *UBIQ10* and *RPS5A* promoters drive high-level and widespread expression of the fused reporter gene. However, we observed SCL23-mVenus fluorescence confined to the lateral organ primordia and not in the meristem center, where endogenous SCL23 is normally expressed (*Figure 5—figure supplement 3C and D*). We found a similar scenario when we tried to overexpress *SHR* or *SCR* in the SAM (*Figure 5—figure supplement 3A and B*), implying tight regulatory controls on GRAS family TFs acting in the different domains in the SAM and preventing excessive protein accumulation. However, excessive accumulation of these TFs was observed outside their normal expression domains (*Figure 5—figure supplement 3A–E*). We propose that SCL23, SHR and SCR are under translational or post-translational control, for example due to the localized presence of miRNAs (*Llave et al., 2002*), and that meristem cells can therefore express only a set maximum amount of these GRAS proteins. We also conclude that SHR controls expression of both SCL23 and SCR in the periphery of the SAM.

# SCL23 acts together with WUS to maintain stem cell homeostasis in the SAM

The *GRAS* family transcription factors HAM1 and HAM2 were shown to physically interact with WUS to promote shoot stem cell proliferation and repress *CLV3* expression in the organizing center (*Zhou et al., 2015*). Since *SCL23* and *WUS* expressions strongly overlap in the organizing center and rib meristem (*Figure 5M and N'*), we asked if SCL23, like HAM1/2, could also interact with WUS. We tested via FRET for in vivo interactions of WUS-FP and SCL23-FP fusions after inducible transient expression in *Nicotiana benthamiana* leaves. SCL23-mVenus interacted with WUS-mCherry with a FRET efficiency similar to the FRET-positive control, a nuclear localized mVenus-mCherry fusion protein (*Figure 5O*). SCL23 showed only a weak tendency to form homomers, assayed by measuring FRET between SCL23-mVenus and SCL23-mCherry (*Figure 5O*). SCL23-mVenus with the unrelated transcription factor AS2-mCherry served as negative control (*Figure 5O*). Given its interaction with WUS and its developmental role in the SAM, we asked if SCL23 expression is also subject to regulation by the *CLV* signaling pathway. We first observed the expression pattern of *SCL23* and *CLV3*. A dual reporter line with *pSCL23:SCL23-YFP* and *pCLV3:NLS-mCherry* showed that *SCL23* and *CLV3* were expressed in largely complementary patterns in the SAM (*Figure 5M and M'*). SCL23 was expressed in the rib meristem, but absent from L1 and L2 layers in the meristem center, while *CLV3* is highly expressed in the L1 and L2 layers and to a lesser level in the L3 layer (*Figure 5M'*). When introduced into the *clv3-9* mutant background, *pSCL23:SCL23-YFP* expression was strongly expanded and extended throughout the enlarged and sometimes fasciated meristems, similar to the altered expression domain of *WUS* in strong *clv*-mutants (*Figure 5—figure supplement 1F–G'*). This supports the hypothesis that *SCL23* expression is, like *WUS*, negatively regulated by *CLV* signaling. To study the developmental role of *SCL23* and *WUS* interaction, we conducted genetic interaction studies. While the SAM of *scl23-2* single mutant was smaller than WT (*Figure 5L*), the null allele *wus-am* had terminated SAM development at an early seedling stage, but continued to initiate new shoot meristems from axillary positions (*Figure 5—figure supplement 4C*). The *wus-am; scl23-2* double mutants showed an enhanced phenotype with earlier arrest of SAM development (*Figure 5—figure supplement 4D*). Hypomorphic *wus-7* mutants form inflorescence meristems similar to WT (*Figure 5—figure supplement 4E–G"*; *Zhou et al., 2015*), while *wus-7; scl23-2* double mutants display premature termination of floral meristems (*Figure 5—figure supplement 4I–I"*). Taken together, the colocalization and interaction between SCL23 and WUS along with the genetic data strongly suggest that SCL23 and WUS function as partners in SAM maintenance.

## Discussion

Understanding the communication and coordination mechanisms between different functional domains within the shoot apical meristem (SAM) is crucial for unraveling the complexities of SAM development and organ formation. In this study, we investigated the role of the *SHR-SCR-SCL23-JKD* gene regulatory network, which was previously shown to control RAM patterning, in the context of the SAM (*Scheres et al., 1995*; *Long et al., 2015a*; *Long et al., 2015b*). We found that all four TFs SHR, SCR, SCL23, and JKD, are expressed in the SAM, exhibiting a combination of complementary and overlapping expression patterns. Mutations in SHR, SCR, or SCL23 resulted in smaller SAMs due to decreased cell proliferation (*Figure 1H*, and *Figure 5L*). Additionally, *shr* and *scr* mutants delayed primordia formation, indicating that *SHR* and *SCR* promote stem cell proliferation within the central zone and primordia initiation in the peripheral zone (*Figure 1D*; *Figure 1—figure supplement 1A'–D'*). These findings are consistent with the well-documented growth-promoting activities of SHR and SCR in leaf and root development (*Dhondt et al., 2010*; *Long et al., 2015a*; *Scheres et al., 1995*). Intriguingly, *jkd* mutants increased SAM size (*Figure 3B–D*), suggesting that *JKD* may restrict shoot growth by interfering with the formation of SHR-SCR complexes, which parallels its previously reported role in roots (*Long et al., 2015b*; *Welch et al., 2007*). Notably, we demonstrated that auxin-triggered primordia initiation in the peripheral zone activates the *SHR* network, which not only promotes cell cycle progression but also establishes communication with the stem cell regulatory system at the center of the meristem through direct interaction with the *WUS-CLV3* feedback loop.

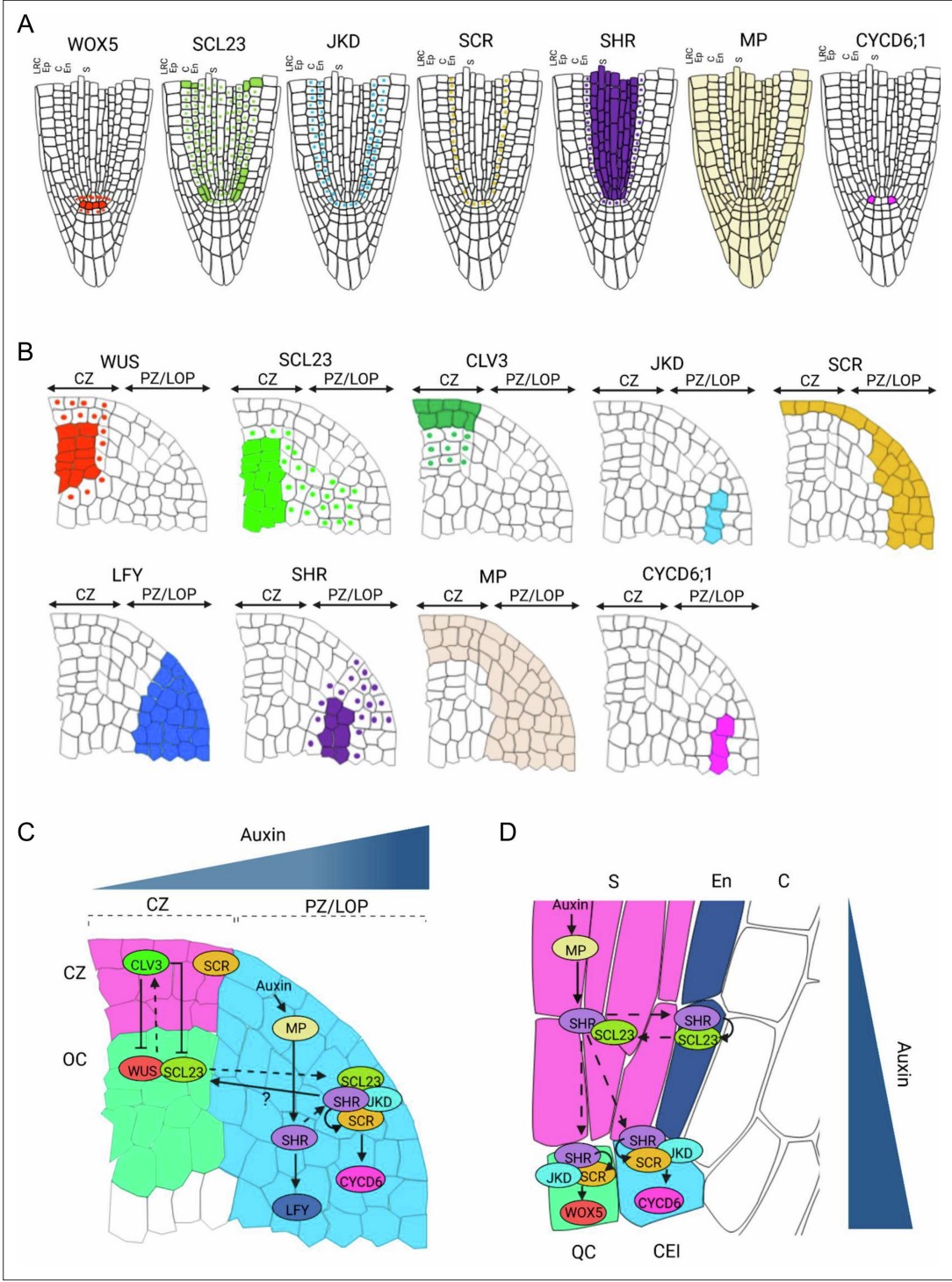

**Figure 6.** Proposed model for SHR-SCR-SCL23-JKD regulatory network function in the SAM and the RAM. (**A** and **B**) Schematic representation of observed expression patterns of WUS, SCL23, CLV3, JKD, SCR, LFY, SHR, MP and CYCD6;1 in the SAM (**A**) and WOX5, SCL23, JKD, SCR, SHR MP and CYCD6;1 in the RAM (**B**). Cells expressing a gene are fully coloured, while presence of mobile proteins is indicated by a coloured nucleus. (**C** and **D**) Schematic molecular models for SHR-SCR-SCL23-JKD gene regulatory network function in the SAM (**C**) and the RAM (**D**), genes with genetic and/or biochemical interactions are indicated. Lines with arrows depict positive regulation, line with arrows with a question mark depict indirect positive regulation, line with bars depict negative regulation and dashed arrows indicate cell-to-cell movement. Overlap between circles describe protein–protein interactions. (**C**) *SHR* transcription is regulated by MP in the PZ/LOP (blue region). In the PZ/LOP, SHR activates *SCR* transcription.

*Figure 6 continued on next page*

*Figure 6 continued*

The SHR-SCR-SCL23-JKD protein complex induces periclinal cell division that leads to the outgrowth of lateral organ primordia through the activation of *CYCD6;1* expression in the PZ/LOP. SHR also regulates the expression of *LFY* in the PZ/LOP, leading to lateral organ initiation. In the organizing center (green region), SCL23 interacts with WUS to maintain stem cell homeostasis in the central zone (CZ) (pink region) where CLV3 is expressed. (**D**) In the RAM, SHR is transcribed in the stele (pink region), and the proteins move outwards to the CEI (blue cell). In the CEI, SHR activates *SCR* and *SCL23* expression, and SHR, SCR and JKD together form a protein complex to induce formative cell divisions via activation of *CYCD6;1* expression. High levels of auxin in the CEI also contribute to *CYCD6;1* activation. In the QC (green box), the SHR-SCR-JKD protein complex positively regulates expression of *WOX5*. CZ = Central zone. OC = Organizing center. PZ/LOP = Peripheral zone/lateral organ primordia, CEI = Cortex/endodermis initial, S=Stele. QC = Quiescent center.

The online version of this article includes the following figure supplement(s) for figure 6:

**Figure supplement 1.** Overlapping expression patterns of PIN1, MP, SHR, SCR, *CYCD6;1*, SCL23 and JKD in the RAM.

## Transcriptional profiles and interdependencies of *SHR*, *SCR*, *SCL23*, and *JKD* in the SAM

*SHR*, *SCR*, *SCL23*, and *JKD* are expressed across all functional domains of the SAM, and mutant analyses emphasized an important role of the *SHR* network in SAM stem cell homeostasis and organogenesis. We found that both SHR and SCL23 proteins are mobile in the SAM, with SCL23 exhibiting hypermobility, similar to its behavior in the RAM (*Figure 2C–D*; *Figure 5A', B', E and F*; *Long et al., 2015a*). This suggests that SCL23 could serve as a communication link between the central zone and peripheral zone, enabling coordination of stem cell homeostasis in the center with lateral organ primordia initiation at the periphery through the conserved *SHR* signaling pathway. Since *SHR* acts as the master regulator within the network, all combinations of *shr*, *scr* and *scl23* mutants displayed the *shr* phenotype (*Figure 1—figure supplement 1A–H'*; *Yoon et al., 2016*). Previous studies on roots revealed that SCR and SCL23 can restrict SHR mobility, likely through complex formation, and that both SCR and SCL23 are transcriptionally regulated by SHR in the RAM (*Long et al., 2015a*). Our studies on SAM development now showed down-regulated expression of *SCR* and *SCL23* in *shr* mutant organ primordia (*Figure 5G and H*; *Figure 2—figure supplement 2F, G and M*), and that the presence of SCR restricted SHR to the nucleus (*Figure 2—figure supplement 2C and 3C inset, and K*). Nuclear retention of SHR due to interaction with SCR could serve to restrict SHR movement. Taken together, these results suggest a regulatory loop where SHR regulates the expression of SCR and SCL23, which in turn restrict SHR mobility. SCR and SCL23 have been shown to function antagonistically in the RAM and redundantly specify endodermal cell fate (*Kim et al., 2017*; *Long et al., 2015a*).

Our data revealed complementary expression patterns of SCL23 and SCR in the SAM's center and their overlap in the primordia where SHR and JKD are also expressed (*Figure 3H1*; *Figure 5C' and D'*; *Figure 3—figure supplement 1H*; *Figure 5—figure supplement 1D and E*). To shed light on the protein interactions within the SHR-SCR-SCL23-JKD network, we investigated the formation of protein complexes in the SAM. Through in vivo FRET-FLIM analysis, we confirmed the physical interaction between SHR, SCR, and JKD in the SAM (*Figure 2H*; 3 N). Moreover, expression analysis of CYCD6;1, a known target of the SHR-SCR-JKD complex in the RAM (*Cui et al., 2007*; *Sozzani et al., 2010*; *Long et al., 2015b*; *Long et al., 2017*), demonstrated its overlap with SHR, SCR, SCL23, and JKD expression in the inner

**Table 1.** Enzymes used in this study.

| Enzyme | Producer |
|---|---|
| BSA1 (ECORI) | Thermo Fisher Scientific, Braunschweig, Germany |
| Pfuµltra High-Fidelity DNA polymerase | Agilent, Santa Clara, USA |
| Phusion High-Fidelity DNA Polymerase | Thermo Fisher Scientific, Braunschweig, Germany |
| T4-Ligase | Thermo Fisher Scientific, Braunschweig, Germany |
| Taq-DNA-Polymerase | Made in the lab according to *Pluthero, 1993* |
| Universal SYBR Green Supermix | Bio-Rad |

**Table 2.** Primers used for cloning.

| Purpose | Primer | Sequence |
| --- | --- | --- |
| | EB-pSHR-F | AAAGGTCTCAACCTGAAGCAGAGCGTGGGGTTTC |
| | EB-pSHR-R | TTTGGTCTCATGTTTTTTAATGAATAAGAAAATGAATAGA AGAAAGGGGG |
| | EB-pSHR-BsaI-site-F | GTTCAAAAGTGGTCCCTTCTCTCTC |
| SHR promoter cloning | EB-pSHR-BsaI-site-R | GAGAGAGAAGGGACCACTTTTGAAC |
| | EB-SHR-CDS-F | AAAGGTCTCAGGCTTAATGGATACTCTCTTTAGACTAGTC AG |
| SHR CDS cloning | EB-SHR-CDS-R | TTTGGTCTCACTGACGTTGGCCGCCACGCACTAG |
| | EB-SCR-CDS-F | AAAGGTCTCAGGCTTAATGGCGGAATCCGGCGATTTC |
| | EB-SCR-CDS-R | TTTGGTCTCACTGAAGAACGAGGCGTCCAAGCTGAAG |
| | EB-SCR-CDS-BsaI-site-1-F | GCCATTATCAGGGACCTTATCC |
| | EB-SCR-CDS-BsaI-site-1-R | GGATAAGGTCCCTGATAATGGC |
| | EB-SCR-CDS-BsaI-site-2-F | GAAAATGGTATCTGCGTTTCAG |
| SCR CDS cloning | EB-SCR-CDS-BsaI-site-2-R | CTGAAACGCAGATACCATTTTC |
| | EB-JKD-CDS-F | AAAGGTCTCAGGCTTAATGCAGATGATTCCAGGAGATCC |
| | EB-JKD-CDS-R | TTTGGTCTCACTGAACCCAATGGAGCAAACCTTGCG |
| | EB-JKD-CDS-BsaI-site-F | GCCCTTGGTGACCTCACTGG |
| JKD CDS cloning | EB-JKD-CDS-BsaI-site-R | CCAGTGAGGTCACCAAGGGC |
| | EB-SCL23-F | AAAGGTCTCAGGCTTAATGACTACAAAACGCATAGACAG |
| SCL23 CDS cloning | EB-SCL23-R | TTTGGTCTCACTGAATCGAACGGCTGAGATTTCC |
| | EB-MP-GG-F | AAAGGTCTCAGGCTTAATGATGGCTTCATTGTCTT |
| MP CDS cloning | EB-MP-GG-R | TTTGGTCTCACTGATGAAACAGAAGTCTTAAGATC |
| | EB-pSHRΔmAuxRE1-F | CTTTGTATCGAGCCAAACGAG |
| | EB-pSHRΔmAuxRE1-R | CTCGTTTGGCTCGATACAAAG |
| | EB-pSHRΔmAuxRE2-F | TTCACATGGCTCTATGTTACTATG |
| | EB-pSHRΔmAuxRE1-R | CATAGTAACATAGAGCCATGTGAA |
| | EB-pSHRΔmAuxRE1-2-F | CTTTGTATCGAGCCAAACGAG |
| | EB-pSHRΔmAuxRE1-2-R | CTCGTTGTGCTCGATACAAAG |
| | EB-pSHRΔmAuxRE2-2-F | ATATTCACATGGGAGTATGTTACTATGTAAATG GTG ACC |
| pSHR site-directed mutagen-esis | EB-pSHRΔmAuxRE2-2-R | GGTCACCATTTACATAGTAACATACTCCCATGTGAATAT |

**Table 3.** Primers used for qRT-PCR.

| Primer name | Sequence |
|---|---|
| EB-RT-SHR-F | GATATCGAGTTTCCGACGGT |
| EB-RT-SHR-R | CGAAGCAAACCCTAAACCAT |
| EB-RT-SCR-F | GTAACCCAAATCTCGGTGCT |
| EB-RT-SCR-R | TTGCTGTTGTGGAGGAGAAG |
| EB-RT-JKD-F | ATCAACCTGGCACTCCAGA |
| EB-RT-JKD-R | GCAGATCTCGCACACGAAT |
| EB-RT-LFY-F | TGATGCTCTCTCCCAAGAAGA |
| EB-RT-LFY-R | CTTGACCTGCGTCCCAGTA |
| EB-SAND-F | AACTCTATGCAGCATTTGATCCACT |
| EB-SAND-R | TGATTGCATATCTTTATCGCCATC |
| EB-TIP41-F | GTGAAAACTGTTGGAGAGAAGCAA |
| EB-TIP41-R | TCAACTGGATACCCTTTCGCA |

**Table 4.** Primers used for genotyping.

| Primer name | Sequence |
|---|---|
| EB-scl23-2-LP | ATGACTACAAAACGCATAGACAG |
| EB-scl23-2-RP | TTTGGTCTCACTGAATCGAACGGC |
| LBb1.3 | ATTTTGCCGATTTCGGAAC |
| mp-S319-LP | CCTGGAAACTGATGAGCTGAC |
| mp-S319-RP | CCTTCTTCACTCATCTGCTGG |
| LBb1.3 | ATTTTGCCGATTTCGGAAC |
| jkd-4-F | GGATGAAAGCAATGCAAAACA |
| jkd-4-R | AATGTCGGGATGATGAACTCC |
| RB | TCAAACAGGATTTTCGCCTGCT |
| scr-4F | CTGCTTCACCTACTGTATGGG |
| scr-4R | GGGTCAGAGGAAGAGGAAGG |
| | Restriction enzyme: *Eco*57M which should not cut in the mutant |
| shr-2-R | AAATCGAACTTGCGAATTCCT |
| shr-2-L | CGCTCAACGAGCTCTCTTCT |
| shr-2ins | CAGCAAGACAAGATGGGTCA |
| wus-7-F | CCGACCAAGAAAGCGGCAACA |
| wus-7-R | AGACGTTCTTGCCCTGAATCTTT |
| | Restriction enzyme: *Xm*NI which should cut in the mutant |

cells of the lateral primordia (**Figure 3H–J**). These findings support the notion that the SHR-SCR-SCL23-JKD complex acts as a positive regulator of cell proliferation in the peripheral zone, possibly through CYCD6;1, leading to lateral organ initiation and outgrowth.

## The *SHR*-network promotes MP and auxin-dependent lateral organ primordia initiation

We found that *SHR* plays a central role in auxin-dependent initiation of lateral organ primordia in the peripheral zone of the SAM. Expression of *SHR* is positively regulated by auxin (**Figure 4A–D**), and two functional AuxRE motifs within the *SHR* promoter are required for *SHR* upregulation (**Figure 4H and M–P**; **Figure 4—figure supplement 5A–J**; **Figure 4—figure supplement 6A–N**; **Figure 4—figure supplement 7A–H**). Although

we have not yet determined which ARFs bind to these motifs in vivo, both expression pattern and *SHR* expression analysis in mutant backgrounds suggest that the ARF MP determines *SHR´s* responses to auxin (**Figure 4I and J**; **Figure 4—figure supplement 3A–D', E, F and I-L**; **Figure 4—figure supplement 5A–J**; **Figure 4—figure supplement 6A–J**; **Figure 4—figure supplement 7A–H**), in line with previous studies which suggested that MP is required for *SHR* expression in the embryo (**Möller et al.,**

**Table 5.** Entry plasmids used in this study.

| Name | Description | Backbone | Reference |
|------|-------------|----------|-----------|
| pSHR | SHR promoter 2.5 Kb upstream from transcription start | pBlunt | This study |
| SHR CDS | SHR coding sequence | pGGC000 | This study |
| SCL23 CDS | SCL23 coding sequence | pGGC000 | This study |
| linker NLS (pGGD007) | NUCLEAR LOCALIZATION SIGNAL | pGGD000 | *Lampropoulos et al., 2013* |
| mVenus (pRD43) | mVenus | pGGD000 | Rebecca Burkhart |
| FLUC | Firefly luciferase | pGGC000 | Greg Denay |
| mVenus (pRD42) | mVenus | pGGC000 | Rebecca Burkhart |
| GUS (pGGC051) | coli ß-GLUCURONIDASE | pGGC000 | *Lampropoulos et al., 2013* |
| MP CDS | MP coding sequence | pGGC000 | This study |
| 35 S promoter (pGGA004) | Cauliflower mosaic virus 35 S promoter | pGGA000 | *Lampropoulos et al., 2013* |
| RPS5A promoter (pGGA012) | RIBOSOMAL PROTEIN 5 A promoter | pGGA000 | *Lampropoulos et al., 2013* |
| UBIQ10 promoter (pGGA006) | UBIQUITIN10 promoter | pGGA000 | *Lampropoulos et al., 2013* |
| d-dummy (pGGD002) | d-dummy | pGGD000 | *Lampropoulos et al., 2013* |
| tCLV3 | CLV3 terminator 1257 bp downstream of transcription stop | pGGE000 | Jenia Schlegel |
| UBQ10 terminator (pGGE009) | UBQ10 terminator | pGGE000 | Lampropoulos et |
| BastaR (pGGF008) | pNOS:BastaR:tNOS | pGGF000 | *Lampropoulos et al., 2013* |
| GR (pRD64) | Hormone-binding domain of the glucocorticoid receptor | pGGD000 | Rebecca Burkhart |
| pCLV3 | CLV3 promoter 1480 bp upstream from transcription start | pGGA000 | Jenia Schlegel |
| mega-element (pGGB002) | Omega- element | pGGB000 | *Lampropoulos et al., 2013* |

*2017*). We found that mutations in *SHR* or *SCR* extended the plastochron, indicating that both genes together with *MP* contribute to lateral organ primordia initiation (*Figure 1D*; *Figure 1—figure supplement 1G*; 9A-B"). Reduction of auxin maxima in *shr* or *scr* mutants (*Figure 1T*), genetic enhancement of *mp* hypomorphs (*Figure 4—figure supplement 4A–D"*) and the interactions with *LFY* expression and function, all support a role for the *SHR*-network in feedback-regulated lateral organ primordia initiation in the peripheral zone, and promotion of lateral organ primordia development.

## A WUS-SCL23 heteromeric complex maintains the SAM stem cell niche

Unlike the other genes in the *SHR* network, *SCL23* exhibited a broad expression pattern in the organizing center, peripheral zone, and lateral organ primordia (*Figure 5B, B', C' and D'*). *SCL23* overlapped with *WUS* expression in the organizing center but was excluded from the central zone (*Figure 5M' and N'*). WUS protein is known to move from the organizing center to the central zone to regulate stem cells and activate *CLV3* expression as part of the negative feedback loop regulation between organizing center and central zone (*Daum et al., 2014*; *Yadav et al., 2011*). We found that WUS and SCL23 engage in heteromeric complexes (*Figure 5O*). SAMs lacking *SCL23* activity remained smaller, indicating that SCL23-WUS complexes promote stem cell activity (*Figure 5L*).

An opposite role to SCL23 had been previously assigned to the HAM TFs, which also belong to the GRAS family. Negative regulation of HAM proteins by a gradient of miRNA171, which originated from the L1 within the SAM, provided a simple mechanism to explain why WUS activates *CLV3* expression only in the central zone at the tip of the SAM (where HAMs are absent), and not in the organizing center itself (where miRNA171 levels are low and WUS interacts with HAMs) (*Zhou et al., 2015*; *Zhou et al., 2018*; *Han et al., 2020a*; *Han et al., 2020b*). Unlike *HAM*, *SCL23* lacks the recognition sequences for regulation by miRNA171. We found that *SCL23* expression is transcriptionally regulated by *CLV3* signaling, thereby confining *SCL23* to the organizing center (*Figure 5M'*; *Figure 5—figure supplement 1F–G'*). Thus, two types of GRAS transcription factors modulate WUS activity: (i) the HAMs, due to their regulation by miRNA171 from the L1, position the domain of WUS activity relative to the SAM surface; (ii) *SCL23* sets the *WUS* domain in relation to the *CLV3* domain and, since *SCL23* is

**Table 6.** Destination plasmids used in this study.

| Name of construct | Promoter | N-Tag | CDS | C-Tag | Terminator | Resistance |
|---|---|---|---|---|---|---|
| pCLV3:SCL23-mVenus | pCLV3 | Ω- element (pGGB002) | SCL23 CDS | mVenus | tCLV3 | BastaR (pGGF008) |
| pRPS5A:SCL23-mVenus | pRPS5A | Ω- element (pGGB002) | SCL23 CDS | mVenus | tUBQ10 (pGGE009) | BastaR (pGGF008) |
| pUBIQ10:SCL23-mVenus | pUBIQ10 | Ω- element (pGGB002) | SCL23 CDS | mVenus | tUBQ10 (pGGE009) | BastaR (pGGF008) |
| pUBIQ10:MP-GR | pUBIQ10 | Ω- element (pGGB002) | MP CDS | GR | tUBQ10 (pGGE009) | BastaR (pGGF008) |
| p35S:GUS | p35S | Ω- element (pGGB002) | GUS | d-dummy (pGGD002) | tUBQ10 (pGGE009) | BastaR (pGGF008) |
| p35S:MP | p35S | Ω- element (pGGB002) | MP CDS | d-dummy (pGGD002) | tUBQ10 (pGGE009) | BastaR (pGGF008) |
| pSHR:mV-NLS | pSHR | Ω- element (pGGB002) | mVenus | linker NLS (pGGD007) | tUBQ10 (pGGE009) | BastaR (pGGF008) |
| pSHRΔmAuxRE1:mV-NLS | pSHRΔmAuxRE1 | Ω- element (pGGB002) | mVenus | linker NLS (pGGD007) | tUBQ10 (pGGE009) | BastaR (pGGF008) |
| pSHRΔmAuxRE2:mV-NLS | pSHRΔmAuxRE2 | Ω- element (pGGB002) | mVenus | linker NLS (pGGD007) | tUBQ10 (pGGE009) | BastaR (pGGF008) |
| pSHRΔmAuxRE1+2:mV-NLS | pSHRΔmAuxRE1+2 | Ω- element (pGGB002) | mVenus | linker NLS (pGGD007) | tUBQ10 (pGGE009) | BastaR (pGGF008) |
| pSHRΔmAuxRE1-2:mV-NLS | pSHRΔmAuxRE1-2 | Ω- element (pGGB002) | mVenus | linker NLS (pGGD007) | tUBQ10 (pGGE009) | BastaR (pGGF008) |
| pSHRΔmAuxRE2-2:mV-NLS | pSHRΔmAuxRE2-2 | Ω- element (pGGB002) | mVenus | linker NLS (pGGD007) | tUBQ10 (pGGE009) | BastaR (pGGF008) |
| pSHRΔmAuxRE1−2+2–2:mV-NLS | pSHRΔmAuxRE1−2+2–2 | Ω- element (pGGB002) | mVenus | linker NLS (pGGD007) | tUBQ10 (pGGE009) | BastaR (pGGF008) |

regulated by the auxin-controlled *SHR* network, in coordination with lateral organ primordia formation in the peripheral zone.

The high mobility of SCL23 in the SAM could allow this protein, in addition to small mobile molecules such as phytohormones, to communicate cell fate and developmental decisions between the organizing center and the peripheral zone. *SHR* would serve as a central integrator, which perceives input from auxin signaling, regulates the expression level and locality of other network components such as *SCR*, *SCL23* and *JKD*, and controls, at least in one instance via *CYCD6;1*, local cell division patterns and growth. The *SHR* regulatory network has been reused in evolutionary times for generating and specifying new cell layers in the RAM, and for maintenance of the quiescent center; the *SHR* network is also in part responsible for coordination of cell fate decisions in the developing vascular system. Although the individual wiring diagrams might differ, the key interactions within this versatile network appear highly conserved (*Figure 6*; *Figure 6—figure supplement 1*).

# Materials and methods
## Plant material and growth conditions
*Arabidopsis thaliana* plants were grown on soil in climate chambers under long day (LD) conditions (16 hr light / 8 hr dark) at 21 °C. Most plants used in this study were in the Columbia (Col-0) background, except for: *scr-4* (WS) (*Fukaki et al., 1998*) and *wus*-7 (*Graf et al., 2010*) in the *Landsberg erecta* (L.er.) background (Tables 4, 8 and 9).

## Plasmid construction and plant transformation
The entry plasmids in this study (Table 5) were generated by gDNA or cDNA sequences that were amplified with Phusion High-Fidelity PCR polymerase using primers described in *Tables 1 and 2*. All

**Table 7.** Destination plasmids used for luciferase assay.

| Name of construct | Promoter | N-Tag | CDS | C-Tag | Terminator | Resistance |
|---|---|---|---|---|---|---|
| pSHR:FLUC | pSHR | Ω- element (pGGB002) | FLUC | d-dummy (pGGD002) | tUBQ10 (pGGE009) | BastaR (pGGF008) |
| pSHRΔmAuxRE1:FLUC | pSHRΔmAuxRE1 | Ω- element (pGGB002) | FLUC | d-dummy (pGGD002) | tUBQ10 (pGGE009) | BastaR (pGGF008) |
| pSHRΔmAuxRE2:FLUC | pSHRΔmAuxRE2 | Ω- element (pGGB002) | FLUC | d-dummy (pGGD002) | tUBQ10 (pGGE009) | BastaR (pGGF008) |
| pSHRΔmAuxRE1+2:FLUC | pSHRΔmAuxRE1+2 | Ω- element (pGGB002) | FLUC | d-dummy (pGGD002) | tUBQ10 (pGGE009) | BastaR (pGGF008) |
| pSHRΔmAuxRE1-2:FLUC | pSHRΔmAuxRE1-2 | Ω- element (pGGB002) | FLUC | d-dummy (pGGD002) | tUBQ10 (pGGE009) | BastaR (pGGF008) |
| pSHRΔmAuxRE2-2:FLUC | pSHRΔmAuxRE2-2 | Ω- element (pGGB002) | FLUC | d-dummy (pGGD002) | tUBQ10 (pGGE009) | BastaR (pGGF008) |
| pSHRΔmAuxRE1−2+2−2:FLUC | pSHRΔmAuxRE1−2+2−2 | Ω- element (pGGB002) | FLUC | d-dummy (pGGD002) | tUBQ10 (pGGE009) | BastaR (pGGF008) |

destination vectors in this study (Table 6 and Table 7) were generated using the GreenGate cloning system (*Lampropoulos et al., 2013*). Entry plasmid containing SHR promoter sequence (2.5 KB) was used as a template to create different promoter mutants using the QuikChange II kit according to manufacturer's protocol (Agilent Technologies). Mutagenic mismatch primers are listed in *Table 2*. all the clones were confirmed by sequencing. *Arabidopsis thaliana* wildtype (Col-0) plants were transformed with *Agrobacterium tumefaciens* (strain GV3101 pMP90 pSoup) containing the respective destination vectors using the floral dipping method (*Clough and Bent, 1998*). Transgenic plants were initially selected on the appropriate antibiotic in the T1 generation. Homozygous T2 plants were identified through confirmation in the T3 generation. Several independent lines were analysed and a representative line was selected for further work. *Nicotiana benthamiana* plants were grown 4–5 weeks in the greenhouse and subsequently used for transient leaf epidermis cell transformation.

### Nicotiana benthamiana infiltration

*Agrobacterium tumefaciens* strains (strain GV3101 pMP90 pSoup) harbouring relevant reporter constructs were cultured overnight with shaking at 28 °C in 5 ml dYT (double Yeast Tryptone, 1.6 % w/v tryptone, 1 % w/v yeast extract, 0.5 % w/v NaCl) with appropriate antibiotics. Cell cultures were adjusted to an optical density ($OD_{600}$) of 0.3 and then harvested by centrifugation at 4000xg for 10 min. The pellet was resuspended in infiltration buffer (5% w/v sucrose, 150 µM acetosyringone, 0.01% v/v Silwet) and incubated for 2 h at 4 °C. For coexpression of two transgenes, the corresponding transformed *A. tumefaciens* were mixed equally. The resuspensions were infiltrated into the abaxial leaf surface of the 3–4 weeks old *N. benthamiana* using a needle-less syringe. Plants transformed with constructs under the control of the 35 S promotor were used for analyses 4 days after infiltration.

**Table 8.** Mutants used in this study.

| Lines | Reference |
|---|---|
| scr-4 | *Fukaki et al., 1998* |
| jkd-4 | *Welch et al., 2007* |
| scr-3 | *Fukaki et al., 1998* |
| shr-2 | *Nakajima et al., 2001* |
| scl23-2 | *Lee et al., 2008* |
| wus-7 | *Graf et al., 2010* |
| wus-am | From Jan lohmann |
| mp-S319 | *Schlereth et al., 2010* |
| mp-B4149 | *Weijers et al., 2005* |
| lfy-12 | *Huala and Sussex, 1992* |
| clv3-9 | *Hobe et al., 2003* |
| cycd6;1 | GABI-Kat line (GK-368E07) *Sozzani et al., 2010* |

### Chemical treatments

For hormone and dexamethasone treatments, plants were grown in soil. For RNA isolation experiments, *pSHR:SHR-YFP*, *pSCR:SCR-YFP* and *pCYCD6;1:GFP* expression analysis following auxin and auxin transport inhibitor treatment were performed by dipping 30-day-old plants inflorescence once in 10 µM (IAA or 2,4 D) once or with 100 µM NPA twice (at 0 hr and 7 hr). For Dexamethasone treatment *pUBIQ1:MP-GR* inflorescence meristems were treated with 10 µM DEX only once and were imaged 5 hr after treatment.

### Promoter mVenus activity in Nicotiana benthamiana

*N. benthamiana* leaves were co-transformed with different *pSHR:mVenus-NLS* and the effector plasmids *p35S:GUS* or *p35S:MP* (Table 6). Four days

**Table 9.** Transgenic lines used in this study.

| Lines | Plant Resistance | Reference |
|---|---|---|
| pSHR:mV-NLS | Basta | This study |
| pSHRΔmAuxRE1:mV-NLS | Basta | This study |
| pSHRΔmAuxRE2:mV-NLS | Basta | This study |
| pSHRΔmAuxRE1+2:mV-NLS | Basta | This study |
| pSHRΔmAuxRE1-2:mV-NLS | Basta | This study |
| pSHRΔmAuxRE2-2:mV-NLS | Basta | This study |
| pSHRΔmAuxRE1−2+2–2:mV-NLS | Basta | This study |
| pCLV3:SCL23-mVenus | Basta | This study |
| pRPS5A:SCL23-mVenus | Basta | This study |
| pUBIQ10:SCL23-mVenus | Basta | This study |
| pUBIQ10:MP-GR | Basta | This study |
| PlaCCI | Kanamycin | *Desvoyes et al., 2020* |
| pSHR:mScarlet-SHR | - | This study |
| pSHR:YFP-SHR | Basta | *Long et al., 2017* |
| pSCR:SCR-RFP | hyg | *Long et al., 2017* |
| pSCR:SCR-YFP | basta | *Long et al., 2017* |
| pSHR:SHR-YFP | Basta | *Long et al., 2017* |
| pJKD:mRFP-YFP | norf | *Long et al., 2017* |
| pJKD:JKD-YFP | Basta | *Long et al., 2017* |
| pSCL23:SCL23-YFP | Kanamycin | *Long et al., 2015a* |
| pSCL23:H2B-YFP | Kanamycin | *Long et al., 2015a* |
| pCYCD6;1:GFP | Basta | *Sozzani et al., 2010* |
| pSCR:H2B-YFP | Kanamycin | *Heidstra et al., 2004* |
| pPIN1:PIN1-GFP | Kanamycin | *Benková et al., 2003* |
| pSHR:nTdTOMATO | - | *Möller et al., 2017* |
| pMP:MP-GFP | Kanamycin | *Schlereth et al., 2010* |
| pLFY:LFY-GFP | kanamycin | *Wu et al., 2003* |
| R2D2 | kanamycin | *Liao et al., 2015* |
| pDR5v2:3xYFP-N7 | Basta | *Heisler et al., 2005* |
| pWUS:3xVenus-NLS/pCLV3-mCherry-NLS | Kanamycin | *Pfeiffer et al., 2016* |
| pCYCD1,1-GFP | - | *Forzani et al., 2014* |
| pCYCD3,2-GFP | - | This study |
| pCYCD3,1-GFP | - | This study |
| pCYCD5,1-GFP | - | This study |
| pCYCD7,1-GFP | - | This study |
| pCYCD2,1-GFP | - | This study |
| pCYCD3,3-GFP | - | *Forzani et al., 2014* |
| pCYCB1;1:CYCB1;1-GFP | Kanamycin | *Ubeda-Tomás et al., 2009* |

after infiltration, the leaves were processed for further analysis using imaging with a Zeiss LSM780 using the same settings for all conditions.

## Luciferase assay in *Nicotiana benthamiana*

The different reporter constructs with firefly luciferase reporter (FLUC) under the control of the different versions of *SHR* promoter (Table 7) were co-infiltrated into *N. benthamiana* leaves together with the effector plasmids *p35S:GUS* or *p35S:MP* (Table 6). Luciferase activities were measured four days after infiltration with the NightOwl luminescence system (Berthold). As substrate for the luciferase reaction, 5 mM D-Luciferin potassium salt solution was used.

## Image acquisition and analysis

All confocal images were obtained by using a Zeiss LSM780 confocal microscope (40 x water immersion objective, Zeiss CPlanApo, NA1.2). For time series analysis, settings were established in the beginning on mock samples and were kept standard during the experiment. Shoot meristems were manually dissected by cutting of the stem, removing the flowers, and were stained with 1 mg/ml DAPI or 5 mM propidium iodide (PI). Individual populations of 2–20 plants were analyzed.

Green fluorescence was excited with an argon laser at 488 nm and emission was detected at 490–530 nm, yellow was excited with an argon laser at 514 nm and emission was detected at 520–550 nm, and red was excited with Diode-pumped solid state (DPSS) lasers at 514 nm and detected at 570–650 nm. PI was excited at 561 nm by DPSS lasers and detected by PMTs at 590–650 nm. DAPI was excited at 405 nm with a laser diode and detected at 410–480 nm.

## FRET-Acceptor-Photobleaching (APB)

*N. benthamiana* leaf epidermal cells were examined using a Zeiss LSM780 confocal microscope (40 x Water immersion objective, Zeiss C-PlanApo, NA1.2). FRET was measured via mVenus fluorescence intensity increase after photobleaching of the acceptor mCherry (*Bleckmann and Simon, 2009*). The percentage change of the mVenus intensity directly before and after bleaching was analyzed as $E_{FRET} = (E_{FRET} = (mVenus_{after} - mVenus_{before})/mVenus_{after} \times 100)$. All photobleaching experiments were performed in the nucleus. The displayed data were obtained from at least three independent experiments.

## FRET-FLIM interaction analysis in the SAM

In vivo FRET-FLIM experiments were measured using a Zeiss LSM780 confocal microscope (40 x Water immersion objective, Zeiss C-PlanApo, NA1.2) equipped with a single-photon counting device (PicoQuant HydraHarp400) and a linear polarized diode laser (LDH-D-C-485).

YFP donor fluorophores was excited with a 485 nm (LDHDC485, 32MHz) pulsed polarized diode laser. Excitation power was adjusted to 1,5µW. Emitted light was separated by a polarizing beam splitter and detected with a band-pass filter (520/35 AHF) by Tau-SPADs (PicoQuant).

Images were acquired with a resolution of 256x256 pixel, zoom 8, a pixel size of 0.1 µm and a dwell time of 12.6µs. For each measurement, 100 frames were taken and the intensity-weighted mean lifetimes $\tau$ (ns) were calculated using PicoQuant SymPhoTime64 software applying a bi-exponential fit. The displayed data were obtained from at least 5 independent experiments.

## Quantitative real time PCR

For qRT-PCR, RNA was isolated from dissected inflorescence meristems using RNeasyMini kit (Qiagen). First strand cDNA was synthesized with 1 µg of RNA cDNA using the Superscript III Kit (Invitrogen). Quantitative real-time PCR was performed with 10-fold diluted cDNA using The SsoAdvancedTM Universal SYBR Green Supermix (Bio-Rad). The mean and standard error were determined using three biological replicates with three technical replicates each. Expression levels were normalized to the references genes TIP41-like (At4g34270) and SAND-domain protein (AT2G28390) (*Czechowski et al., 2005*). Primers used are listed in *Table 3*. Calculation of the relative expression was performed according to *Pfaffl, 2001*.

## Phenotypic analyses

Rosettes and plants were analyzed by taking photographs using Canon EOS400D camera with an EF-S 60 mm Canon ZOOM lens of plants growing on soil. Inflorescences were analyzed by taking pictures

using stereo microscope (Nikon SMZ25). The inflorescence plastochron was obtained by calculating the average time separating the emergence of 2 successive flower above stage 15 emerging after plant bolting. For SAM area measurement, 30DAG plants growing under LD conditions were used. The primary inflorescence was dissected and imaged using LSM780. The SAM area measurement was done by Fiji.

## Software

Microsoft Word, Excel, and PowerPoint software was used to organize experimental data. Images were analyzed and processed with ImageJ v 1.53 c (*Schneider et al., 2012*) and Carl Zeiss ZEN2011. All images were adjusted in "Brightness and Contrast". VectorNTI (InvitrogenTM) was used for vector maps and sequence analysis. Databank gene research were performed on The *Arabidopsis* Information Resource (TAIR), (http://www.arabidopsis.org/). Indigo was used for the luciferase assay imaging. Statistical analyses and box plots were realized with GraphPad Prism v 8. For visualization. Using the open-source software MorphoGraphX (MGX) software (https://morphographx.org/) (*Barbier de Reuille et al., 2015*) the surface of the meristem was extracted and the PI signal of the cell wall from layer1 (L1) was projected and used for segmentation of the images to quantify number and size of cells. Cells were segmented manually. MorphographX analysis was performed according to standards defined in the user manual (*Barbier de Reuille et al., 2015*). The visualisation and counting of nuclei expressing PlaCCI (*Desvoyes et al., 2020*) was done with Imaris (version 9.1.2, Bitplane, Oxford Instruments plc). Ratios for R2D2 were calculated as described previously (*Bhatia et al., 2016*). All Statistical analyses and data plotting were realized with GraphPad Prism v 8. All images for an experimental set were captured under identical microscope settings.

## Acknowledgements

We thank Jan Lohmann, Dolf Weijers, Francois Parcy and Doris Wagner for providing seeds. We are also grateful to the CAi team at HHU, especially Sebastian Hänsch for support with imaging methods, and Patrick Blümke, Jan Maika, Jenia Schlegel, Maike Breiden, Meik Thiele, Karine Gustavo Pinto, Svenja Augustin, Carin Theres, Silke Winters, Vivien Strotmann, Rebecca Drisch, Madhumita Narasimhan, Vicky Howe, Pavithran Narayanan and Cornelia Gieseler for diverse contributions. Research in the lab of Rüdiger Simon is supported by the DFG through the Cluster of Excellence on Plant Sciences (CEPLAS, EXC2048), Next-Plant, SFB1208, CRC1208 and RTG CSCS. Research in the lab of Crisanto Gutierrez is supported by the Grant ERC-2018-AdG_833617 (European Union, H2020). Grant RTI2018-094793-B-I00 (from Spanish Ministry of Science and Innovation, and FEDER).

## Additional information

### Funding

| Funder | Grant reference number | Author |
|---|---|---|
| Deutsche Forschungsgemeinschaft | CSCS | Rüdiger GW Simon |
| Deutsche Forschungsgemeinschaft | EXC2048 | Elmehdi Bahafid<br>Imke Bradtmöller<br>Ann M Thies<br>Thi TON Nguyen<br>Yvonne Stahl<br>Rüdiger GW Simon |
| Deutsche Forschungsgemeinschaft | CRC1208 | Rüdiger GW Simon |
| European Union | ERC-2018-AdG_833617 | Crisanto Gutierrez<br>Bénédicte Desvoyes |
| Spanish Ministry of Science and Innovation | RTI2018-094793-B-I00 | Crisanto Gutierrez<br>Bénédicte Desvoyes |

| Funder | Grant reference number | Author |
|---|---|---|

The funders had no role in study design, data collection and interpretation, or the decision to submit the work for publication.

## Author contributions

Elmehdi Bahafid, Conceptualization, Investigation, Writing – original draft, Writing – review and editing; Imke Bradtmöller, Ann M Thies, Thi TON Nguyen, Investigation; Crisanto Gutierrez, Bénédicte Desvoyes, Resources, Writing – original draft; Yvonne Stahl, Resources, Methodology, Writing – original draft; Ikram Blilou, Conceptualization, Resources, Supervision, Investigation, Methodology, Writing – original draft; Rüdiger GW Simon, Conceptualization, Resources, Supervision, Funding acquisition, Investigation, Writing – original draft, Project administration, Writing – review and editing

## Author ORCIDs

Elmehdi Bahafid (ID) http://orcid.org/0000-0002-2678-6246
Crisanto Gutierrez (ID) https://orcid.org/0000-0001-8905-8222
Rüdiger GW Simon (ID) http://orcid.org/0000-0002-1317-7716

## Decision letter and Author response

Decision letter https://doi.org/10.7554/eLife.83334.sa1
Author response https://doi.org/10.7554/eLife.83334.sa2

# Additional files

## Supplementary files

• MDAR checklist

## Data availability

Original microscopy and image analysis data referenced in the manuscript are accessible through BioStudies at the following link: https://www.ebi.ac.uk/biostudies/bioimages/studies/S-BIAD903.

The following dataset was generated:

| Author(s) | Year | Dataset title | Dataset URL | Database and Identifier |
|---|---|---|---|---|
| Bahafid E | 2023 | The Arabidopsis SHORTROOT network coordinates shoot apical meristem development with auxin dependent lateral organ initiation | https://www.ebi.ac.uk/biostudies/bioimages/studies/S-BIAD903 | BioStudies, S-BIAD903 |

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
