## [Editor Report]

This is a valuable study of Arabidopsis shoot apical meristem maintenance and organ initiation, with a focus on how SHR, SCR, JKD, and SCL23, four transcription factors initially characterized in root meristems, are deployed in a different context. The imaging, genetics and FRET-FLIM evidence supporting the claims of the authors is compelling. The work will be of interest and importance for plant developmental biologists.

---

## [Decision Letter]

**Decision letter after peer review:**

Thank you for submitting your article "The Arabidopsis SHORTROOT network coordinates shoot apical meristem development with auxin dependent lateral organ initiation" for consideration by *eLife*. Your article has been reviewed by 3 peer reviewers, one of whom is a member of our Board of Reviewing Editors, and the evaluation has been overseen by Jürgen Kleine-Vehn as the Senior Editor. The reviewers have opted to remain anonymous.

Essential revisions:

1) Although the amount of data included is impressive, the current length and organization of the manuscript obscure the story. Making clearer connections between sections, defining a clear message, streamlining (removing) peripheral information and reducing unnecessary background information are required. Specific suggestions about how to cut or combine material made by individual reviewers are excellent guidelines. [e.g., the first two result sections as well as figure 1 and 2 can be merged, auxin-related results can be moved together, and the LFY results would fit better if they were condensed and moved to the "MONOPTEROS regulates SHR and SCR expression in the SAM" section, possibly after line 372 as an example of MP -mediated regulation of organ initiation].

2) As this is a highly dense study, the overview figure, collecting the results and situating them in a broader perspective is needed. However, the model figure (13) was not very helpful in understanding the regulator relationships among the GRAS factors and the differences/commonalities between the RAM and SAM. One potential solution would be to place the diagram of RAM factors (Figure S18B) beside the SAM diagram and to discuss what is and isn't the same and if the differences map on to different things the meristems do, or to the presence of a unique TF. Additional detail should be added. For example, the expression pattern of SCR and SCL23 seems not to overlap and yet these proteins are drawn in the same complex. The inclusion of SCL23 in the core complex with SHR and SCR is also inconsistent with LOF phenotypes reported as being additive (line 460). Indicating which interactions are experimentally validated in the SAM and which are putative interactions, would avoid ambiguous interpretation. The authors could also add different shades for the domains where the protein moves so the reader can easily distinguish between transcriptional and translational expression.

3) Several issues need to be resolved with the JKD model. One reviewer notes that the SHR-SCR interaction seems to happen in different cells than the SCR-JKD interaction. Are representative images and/ or protein-protein imaging data that could support the conclusion of the authors that SHR-SCR-JKD are localized in the same cell type and that is where they interact? A second reviewer was not clear on the model for JKD influencing the SCR and SHR dimers and activities in the meristem. PlaCCI expression appears to be measured in epidermis, but JKD isn't there, so is the presumed longer G1 phase indirect and due to the proposed sequestering of SHR-SCR? Given the lower levels of JKD relative to SHR and SCR, is this a reasonable hypothesis? Also starting on line 276, it was not clear how the data for JKD-SCR indicate the behavior with SHR-SCR. Lifetime measurements seem to go down the same about for FRET data in 6H and 4E. It was not clear how this assay could distinguish between binary and ternary complexes of SHR-SCR-JKD.

4) In general, the conclusion statements at the end of the paragraphs oversimplify and overstate the evidence. For example, line 165-7 "We conclude that SHR and SCR promote cell cycle progression in the meristem by controlling the G1 phase, which could account for the observed reduction in SAM size and delay in organ initiation in the PZ of shr and scr mutants." Or line 375-6 "We conclude that MP acts through SHR to promote lateral organ initiation". More experimental evidence would be required to make these firm conclusions, but an alternative would be substituting the phrase "our data are consistent with a model that" for "we conclude".

5) Some conclusions are not supported by data. For example, the authors conclude that SHR and cofactors drive cell proliferation through CYCD6, but their evidence is that "We found that the number of cell layers in the L3 of lateral organ primordia is strongly reduced in shr-2 mutants compared to WT." It would be essential to examine the SAM of the cycd6;1 mutant to make this conclusion.

*Reviewer #1 (Recommendations for the authors):*

This long manuscript contains many well done experiments, but the overall story seems underwhelming relative to the data, and could use some streamlining in the text as well as better set up of expectations for when SAM and RAM would be expected to use the same factors in the same way vs. when it would be surprising if they did. It felt like a missed opportunity to highlight how analysis of the GRAS TF network in another tissue extends our understanding of the regulatory logic of these factors and/or plants.

1) The authors say "In the root meristem, CYCD6;1 specifically controls periclinal cell divisions and the formation of new tissue layers (Long et al., 2015b). We found that the number of cell layers in the L3 of lateral organ primordia is strongly reduced in shr-2 mutants compared to WT." Yet the authors didn't examine the SAM of the cycd6;1 mutant. I think it would be essential to do so to make this conclusion. My own observation is that the cycd6;1 phenotype is too mild for it to be placed at the center of the root response, and there has been some over interpretation of CYCD6 function; it may just be a very handy expression marker.

2) In general, the conclusion statements at the end of the paragraphs oversimplify and overstate the evidence. For example, line 165-7 "We conclude that SHR and SCR promote cell cycle progression in the meristem by controlling the G1 phase, which could account for the observed reduction in SAM size and delay in organ initiation in the PZ of shr and scr mutants." Or line 375-6 "We conclude that MP acts through SHR to promote lateral organ initiation". More experimental evidence would be required to make these firm conclusions, but an alternative would be substituting the phrase "our data are consistent with a model that" for "we conclude".

3) Figure 10. The figure shows that AREs are necessary for SHR expression, but I think the point that MP could bind these would be made better if some of the data from Figure S8B and S9B were included in this main figure.

4) I did not find the model figure (13) very helpful in understanding the regulator relationships among the GRAS factors and the differences/commonalities between the RAM and SAM. One potential solution would be to place the diagram of RAM factors (Figure S18B) beside the SAM diagram and to discuss what is and isn't the same and if the differences map on to different things the meristems do, or to the presence of a unique TF. I was also confused by the inclusion of SCL23 in the core complex with SHR and SCR because the LOF phenotypes were reported as being additive (line 460).

5) I was confused by some of the explanations of the JKD experiments and results. It was not clear to me what the model is for JKD influencing the SCR and SHR dimers and activities in the meristem. PlaCCI expression appears to be measured in epidermis, but JKD isn't there, so I guess the presumed longer G1 phase must be indirect and potentially due to the proposed sequestering of SHR-SCR. But given the lower levels of JKD relative to SHR and SCR, is this a reasonable hypothesis? Also starting on line 276, it was not clear to me how the data for JKD-SCR indicate the behavior with SHR-SCR. Lifetime measurements seem to go down the same about for FRET data in 6H and 4E. I could not see how this assay could distinguish between binary and ternary complexes of SHR-SCR-JKD.

*Reviewer #2 (Recommendations for the authors):*

1. Whereas SHR and SCR mutants have strong shoot development phenotypes, shoot development in mutants of JKD and SCL23 are relatively normal. Can the authors use a dominant-negative version of SCL23, such as fusing with EAR to test the roles of scl23 in the CZ?

2. Why SHR and SCR mutants have much smaller SAM (than SCL23 mutants), although they do not express in the SAM (as SCL23)?

*Reviewer #3 (Recommendations for the authors):*

Even though the authors explained the results carefully and critically, the manuscript would benefit from shortening and restructuring of the results. For example, the first two result sections as well as figure 1 and 2 can be merged, auxin-related results can be moved together, and the LFY results would fit better if they were condensed and moved to the "MONOPTEROS regulates SHR and SCR expression in the SAM" section, possibly after line 372 as an example of MP -mediated regulation of organ initiation.

The authors use a very systematic approach, where they show that SHR and SCR expression overlaps and the SAM is reduced in the mutant due to fewer cells. They show that SHR and SCR are interacting, which seems to localize at P1. Given the known role of JKD in the root, the authors continue to look into JKD and show the interaction between SCR-JKD. However, the reviewer is questioning whether JKD and SHR interact in the SAM. The SHR-SCR interaction seems to happen in different cells than the SCR-JKD interaction. Are representative images and/ or protein-protein imaging data that could support the conclusion of the authors that SHR-SCR-JKD are localized in the same cell type and that is where they interact?

As this is a highly dense study, the overview figure, collecting the results and situating them in a broader perspective is needed. However, the presented overview figure 13 would greatly improve if additional detail can be added. For example, the expression pattern of SCR and SCL23 seems not to overlap and yet these proteins are drawn in the same complex. Thus, indicating which interactions are experimentally validated in the SAM and which are putative interactions, would avoid ambiguous interpretation. The authors could also add different shades for the domains where the protein moves so the reader can easily distinguish between transcriptional and translational expression.

The authors state that SCR does not homodimerize in the root, however, specifically in the QC SCR does form dimers as shown by Clark et al. 2020. Please rephrase this statement in the results or elaborate on the sensitivity of FRET-FLIM, which could miss the yet shown 5% SCR-SCR dimer formation.

---

## [Author Response]

Essential revisions:1) Although the amount of data included is impressive, the current length and organization of the manuscript obscure the story. Making clearer connections between sections, defining a clear message, streamlining (removing) peripheral information and reducing unnecessary background information are required. Specific suggestions about how to cut or combine material made by individual reviewers are excellent guidelines. [e.g., the first two result sections as well as figure 1 and 2 can be merged, auxin-related results can be moved together, and the LFY results would fit better if they were condensed and moved to the "MONOPTEROS regulates SHR and SCR expression in the SAM" section, possibly after line 372 as an example of MP -mediated regulation of organ initiation].

We greatly appreciate all the suggestions provided. We have incorporated them into the manuscript, leading us to restructure both the figures and the results and Discussion sections. The manuscript has also been shortened, and although new data were added, we reduced the total number of figures. We hope that we thereby addressed all the reviewers and editors concerns, and submit here a vastly improved version.

2) As this is a highly dense study, the overview figure, collecting the results and situating them in a broader perspective is needed. However, the model figure (13) was not very helpful in understanding the regulator relationships among the GRAS factors and the differences/commonalities between the RAM and SAM. One potential solution would be to place the diagram of RAM factors (Figure S18B) beside the SAM diagram and to discuss what is and isn't the same and if the differences map on to different things the meristems do, or to the presence of a unique TF.

We have changed the model based on your suggestions, and present a new summary figure that gives a better overview of the results. It also highlights the relationships and interactions between GRAS factors and their partners, and displays commonalities and important differences RAM and SAM.

Additional detail should be added. For example, the expression pattern of SCR and SCL23 seems not to overlap and yet these proteins are drawn in the same complex.

Regarding the expression patterns of SCR and SCL23, it's important to note that their expression does overlap in the primordia. This can be seen in Figure 5C-D' and Figure 5-FigSuppl1D, indicated by the arrowheads. In addition, we conclude that SCL23 protein is very mobile, which was already noted earlier by others when studying SCL23 function in the root system. We also found that while SCL23 promoter activity is confined to the organising centre of the SAM, the SCL23 fusion proteins are found in a much larger domain. We deduce from this observation (described in the text) that SCL23 protein can serve to communicate cell fate decisions between CZ and PZ.

The inclusion of SCL23 in the core complex with SHR and SCR is also inconsistent with LOF phenotypes reported as being additive (line 460).

We appreciate the opportunity to clarify these points. In our study, we have presented evidence for the interaction between SHR and SCR in the SAM. Because SHR, SCR and SCL23 proteins are all found in specific regions of the PZ, we deduce that protein complexes comprising SHR, SCR and SCL23 can form there. This is illustrated in Figure 5C-D' and Figure 5-FigSuppl1D (arrowheads). It has been shown previously that in RAM, SHR, SCR and SCL23 proteins can interact *(*Long et al., 2015a). In our manuscript, we have also taken into account the functional aspects of this complex. Our results indicate that SHR plays a pivotal role as the core component in this signaling pathway. This conclusion is supported by our demonstration that *shr* mutants exhibit epistatic effects over *scl23* mutants in double mutant combinations, as shown in Figure 5-FigSuppl2A-H'. This observation aligns with similar findings presented by Yoon et al., 2016. We hope this clarifies this point!

Indicating which interactions are experimentally validated in the SAM and which are putative interactions, would avoid ambiguous interpretation. The authors could also add different shades for the domains where the protein moves so the reader can easily distinguish between transcriptional and translational expression.

We have incorporated distinct labeling to allow readers to easily differentiate between transcriptional and translational reporter lines, and therefore gene expression and presence of the (mobile) protein (Fig6).

3) Several issues need to be resolved with the JKD model. One reviewer notes that the SHR-SCR interaction seems to happen in different cells than the SCR-JKD interaction. Are representative images and/ or protein-protein imaging data that could support the conclusion of the authors that SHR-SCR-JKD are localized in the same cell type and that is where they interact? A second reviewer was not clear on the model for JKD influencing the SCR and SHR dimers and activities in the meristem.

SHR, SCR, and JKD proteins are all present within the same overlapping domain, specifically in the lateral organ primordia, which is now illustrated better in Figures 3H and 3I. In this domain, JKD interacts with SCR (Fig3N), and in the same domain, SHR also interacts with SCR (Fig2F and Fig2H).

PlaCCI expression appears to be measured in epidermis, but JKD isn't there, so is the presumed longer G1 phase indirect and due to the proposed sequestering of SHR-SCR?

It is possible that JKD operates in a manner similar to its role in the root, exerting control over SHR mobility and prompting additional ectopic cell divisions. Addressing the cell cycle reporter aspect, we examined the expression of pCDT1a:CDT1a-eCFP, pHTR13:pHTR13-mCherry, and pCYCB1;1:NCYCB1;1-YFP in cells across the entire meristem excluding lateral organ primordia. To achieve this, we employed nucleus segmentation through Imaris, as detailed in the material and methods section. Figure 1-FigSuppl1P shows a visual representation of our measurements.

Given the lower levels of JKD relative to SHR and SCR, is this a reasonable hypothesis? Also starting on line 276, it was not clear how the data for JKD-SCR indicate the behavior with SHR-SCR. Lifetime measurements seem to go down the same about for FRET data in 6H and 4E. It was not clear how this assay could distinguish between binary and ternary complexes of SHR-SCR-JKD.

Our FRET assays can only resolve binary interactions, and not the formation of putative ternary complexes. However, our results demonstrate JKD's interaction with SCR within the same cells where SHR interacts with SCR. Based on these findings, we have inferred that JKD likely forms a complex in conjunction with both SCR and SHR. This assumption is based also on the consistency of our observations and the results published by Long et al., 2017.

4) In general, the conclusion statements at the end of the paragraphs oversimplify and overstate the evidence. For example, line 165-7 "We conclude that SHR and SCR promote cell cycle progression in the meristem by controlling the G1 phase, which could account for the observed reduction in SAM size and delay in organ initiation in the PZ of shr and scr mutants." Or line 375-6 "We conclude that MP acts through SHR to promote lateral organ initiation". More experimental evidence would be required to make these firm conclusions, but an alternative would be substituting the phrase "our data are consistent with a model that" for "we conclude".

We have revised the conclusion statements throughout the manuscript and have made the necessary modifications.

5) Some conclusions are not supported by data. For example, the authors conclude that SHR and cofactors drive cell proliferation through CYCD6, but their evidence is that "We found that the number of cell layers in the L3 of lateral organ primordia is strongly reduced in shr-2 mutants compared to WT." It would be essential to examine the SAM of the cycd6;1 mutant to make this conclusion.

We concluded that SHR and cofactors drive cell proliferation through CYCD6;1, substantiated by the significant reduction in pCYCD6;1-GFP expression within the lateral organ primordia of the *shr-2* mutant. This decrease in expression corresponds with the reduction in the number of cell layers within the L3 of the lateral organ primordia in *shr-2* mutants, compared to wild-type. To further support this conclusion, we have added new data by analyzing the meristem of the *cycd6;1* mutant. Our findings reveal a small, but significant reduction in both meristem size and the number of cell layers in the L3, relative to the wild type, as depicted in Figure 4-FigSuppl2 7I-N. Collectively, these findings underscore our assertion that the SHR regulatory network plays a role in activating CYCD6;1 expression, thereby promoting cell division within the lateral organ primordia.

Reviewer #1 (Recommendations for the authors):This long manuscript contains many well done experiments, but the overall story seems underwhelming relative to the data, and could use some streamlining in the text as well as better set up of expectations for when SAM and RAM would be expected to use the same factors in the same way vs. when it would be surprising if they did. It felt like a missed opportunity to highlight how analysis of the GRAS TF network in another tissue extends our understanding of the regulatory logic of these factors and/or plants.

We have streamlined the text and placed more emphasis on establishing clear expectations for the usage of common factors between SAM and RAM, addressing similarities and differences.

1) The authors say "In the root meristem, CYCD6;1 specifically controls periclinal cell divisions and the formation of new tissue layers (Long et al., 2015b). We found that the number of cell layers in the L3 of lateral organ primordia is strongly reduced in shr-2 mutants compared to WT." Yet the authors didn't examine the SAM of the cycd6;1 mutant. I think it would be essential to do so to make this conclusion. My own observation is that the cycd6;1 phenotype is too mild for it to be placed at the center of the root response, and there has been some over interpretation of CYCD6 function; it may just be a very handy expression marker.

see response above.

2) In general, the conclusion statements at the end of the paragraphs oversimplify and overstate the evidence. For example, line 165-7 "We conclude that SHR and SCR promote cell cycle progression in the meristem by controlling the G1 phase, which could account for the observed reduction in SAM size and delay in organ initiation in the PZ of shr and scr mutants." Or line 375-6 "We conclude that MP acts through SHR to promote lateral organ initiation". More experimental evidence would be required to make these firm conclusions, but an alternative would be substituting the phrase "our data are consistent with a model that" for "we conclude".

We have reviewed the highlighted sections and have made necessary adjustments to ensure that our conclusions accurately reflect the level of evidence presented.

3) Figure 10. The figure shows that AREs are necessary for SHR expression, but I think the point that MP could bind these would be made better if some of the data from Figure S8B and S9B were included in this main figure.

We appreciate your suggestion regarding Figure 10. While we acknowledge that including data from Figure S8B and S9B could enhance the point about MP potentially binding to AREs, we opted not to make this modification in order to maintain clarity and avoid overcrowding the main figures. Our intention was to create a balance between providing comprehensive information and ensuring figure legibility. However, we've made sure that the relevant data and information from Figure S8B and S9B are well-referenced in the text, allowing readers to easily access and understand the supporting evidence.

4) I did not find the model figure (13) very helpful in understanding the regulator relationships among the GRAS factors and the differences/commonalities between the RAM and SAM. One potential solution would be to place the diagram of RAM factors (Figure S18B) beside the SAM diagram and to discuss what is and isn't the same and if the differences map on to different things the meristems do, or to the presence of a unique TF.

We have changed the model based on your suggestions, and present a new summary figure that gives a better overview of the results. It also highlights the relationships and interactions between GRAS factors and their partners, and displays commonalities and important differences RAM and SAM.

I was also confused by the inclusion of SCL23 in the core complex with SHR and SCR because the LOF phenotypes were reported as being additive (line 460).

Our study establishes that SHR and SCR interact in the SAM, where the overlapping expression patterns of SHR, SCR, and SCL23 are observed, as depicted in Figure 5C-D' and Figure 5-FigSuppl1D (arrowheads). Integration of SCL23 in a complex with SHR and SCR has been experimentally validated in vivo by Long et al., 2015a. Additionally, the downregulation of SCL23 expression in the *shr* mutant provides further support for its involvement in the same pathway, acting downstream of SHR. With this in mind, the additive phenotype indicates that complexes between SCL23, SHR and SCR also comprise other partners, with partially redundant functions.

5) I was confused by some of the explanations of the JKD experiments and results. It was not clear to me what the model is for JKD influencing the SCR and SHR dimers and activities in the meristem. PlaCCI expression appears to be measured in epidermis, but JKD isn't there, so I guess the presumed longer G1 phase must be indirect and potentially due to the proposed sequestering of SHR-SCR. But given the lower levels of JKD relative to SHR and SCR, is this a reasonable hypothesis? Also starting on line 276, it was not clear to me how the data for JKD-SCR indicate the behavior with SHR-SCR. Lifetime measurements seem to go down the same about for FRET data in 6H and 4E. I could not see how this assay could distinguish between binary and ternary complexes of SHR-SCR-JKD.

It is important to note that SHR, SCR, and JKD are all expressed within the same overlapping domain, as illustrated in Figure 3H and I.

Addressing the cell cycle reporter aspect, we examined PlaCCI expression (pCDT1a:CDT1a-eCFP, pHTR13:pHTR13-mCherry, and pCYCB1;1:NCYCB1;1-YFP) in cells across the entire meristem, not only in the epidermis. To achieve this, we employed nucleus segmentation through Imaris, as detailed in the material and methods section. See Figure 1-FigSuppl1P for a visual representation of our measurements. PlaCCI was studied in wildtype, *shr* and *scr* mutants, but not in *jkd* mutants, so we cannot speculate on a general role of JKD in cell cycle regulation beyond CYCD6;1.

Turning to the complex formation involving SHR, SCR, and JKD, we assayed FRET-FLIM within the shared domain of these factors. Such FRET assays do not allow to distinguish between binary and tertiary complexes, so the exact composition and stoichiometry of the complexes is not known. Nevertheless, we found that JKD interacts with SCR within the same cellular context where SHR also engages with SCR, as depicted in Figure 2H and 3N, (and similar interactions were found for SHR, SCR and JKD in the RAM, reported in Long et al., 2017).

Reviewer #2 (Recommendations for the authors):1. Whereas SHR and SCR mutants have strong shoot development phenotypes, shoot development in mutants of JKD and SCL23 are relatively normal. Can the authors use a dominant-negative version of SCL23, such as fusing with EAR to test the roles of scl23 in the CZ?

We have described distinct phenotypes associated with both *jkd* and *scl23* mutants in the manuscript: SAM sizes of *jkd-4* mutants are increased compared to the wild type (WT), indicating a significant impact on shoot development. We observed decreased SAM sizes in *scl23* mutants compared to WT, underlining the role of SCL23 in shoot development.

2. Why SHR and SCR mutants have much smaller SAM (than SCL23 mutants), although they do not express in the SAM (as SCL23)?

The differences in SAM sizes between SHR, SCR, and SCL23 mutants can be attributed to their distinct expression patterns and potential roles in the meristem. SCR is expressed in the peripheral zone, in the L1 layer in the center of the meristem and in lateral organ primordia, while SHR is primarily expressed in the lateral organ primordia but absent from the meristem's center. On the other hand, SCL23 is expressed in both the center of the meristem and lateral organ primordia. We propose that these genes are interconnected within the same signaling system. The expression of genes in the lateral organ primordia and those in the center of the meristem suggests a coordinated communication mechanism. This interaction contributes to the coordination of primordia initiation at the meristem's periphery with cell proliferation in the meristem's center. While the exact mechanisms need further investigation, our observations suggest that these genes play roles in regulating different aspects of meristem development that collectively impact SAM size.

Reviewer #3 (Recommendations for the authors):Even though the authors explained the results carefully and critically, the manuscript would benefit from shortening and restructuring of the results. For example, the first two result sections as well as figure 1 and 2 can be merged, auxin-related results can be moved together, and the LFY results would fit better if they were condensed and moved to the "MONOPTEROS regulates SHR and SCR expression in the SAM" section, possibly after line 372 as an example of MP -mediated regulation of organ initiation.

We shortened and restructured the manuscript.

The authors use a very systematic approach, where they show that SHR and SCR expression overlaps and the SAM is reduced in the mutant due to fewer cells. They show that SHR and SCR are interacting, which seems to localize at P1. Given the known role of JKD in the root, the authors continue to look into JKD and show the interaction between SCR-JKD. However, the reviewer is questioning whether JKD and SHR interact in the SAM. The SHR-SCR interaction seems to happen in different cells than the SCR-JKD interaction. Are representative images and/ or protein-protein imaging data that could support the conclusion of the authors that SHR-SCR-JKD are localized in the same cell type and that is where they interact?

Note that the overall expression patterns of SHR, SCR, and JKD are different, but they are also expressed within an overlapping domain, see for example Fig3H,I; here, the three proteins (and potentially other partners) can interact in multimeric complexes. We measured FRET-FLIM in this domain, and found that they interact.

As this is a highly dense study, the overview figure, collecting the results and situating them in a broader perspective is needed. However, the presented overview figure 13 would greatly improve if additional detail can be added. For example, the expression pattern of SCR and SCL23 seems not to overlap and yet these proteins are drawn in the same complex. Thus, indicating which interactions are experimentally validated in the SAM and which are putative interactions, would avoid ambiguous interpretation. The authors could also add different shades for the domains where the protein moves so the reader can easily distinguish between transcriptional and translational expression.

We have taken this comment and similar comments by the other reviewers into account and redesigned the figure.

The authors state that SCR does not homodimerize in the root, however, specifically in the QC SCR does form dimers as shown by Clark et al. 2020. Please rephrase this statement in the results or elaborate on the sensitivity of FRET-FLIM, which could miss the yet shown 5% SCR-SCR dimer formation.

The sensitivity of our FRET experiments would not allow to detect a minor fraction of homodimers in vivo, we have therefore changed the text accordingly.